# Modeling atmospheric sulfate oxidation chemistry via the oxygen isotope anomaly using the Community Multiscale Air Quality Model (CMAQ)

Huan Fang[1], Wendell Walters[1]

[1]Department of Chemistry and Biochemistry, University of South Carolina, SC, 631 Sumter Street Columbia, SC 29208, United States

*Correspondence to*: Wendell Walters (wendellw@mailbox.sc.edu)

**Abstract.** Atmospheric sulfate formation influences climate and air quality, yet its chemical pathways remain difficult to constrain. This study utilizes the oxygen isotope anomaly ($\Delta^{17}O$) of sulfate aerosol ($ASO_4$) as a tracer to distinguish formation processes. This work presents a simulation of $\Delta^{17}O(ASO_4)$ within the contiguous United States, conducted over full annual cycles, which enables the quantification of seasonal and spatial patterns of sulfate oxidation pathways and their response to major emission reductions, for the first time at this scale and temporal coverage. In 2019, $\Delta^{17}O(ASO_4)$ values were predicted to be below 1‰ in the Gulf Coast, indicating acidic, $ASO_4$-rich conditions dominated by $S(IV) + H_2O_2$ oxidation, while values above 2‰ in the West suggested less acidic conditions, leading to enhanced $ASO_4$ production via $S(IV) + O_3$ oxidation. Peak $\Delta^{17}O(ASO_4)$ values of ~4.5‰ in April across the Western US reflected $O_3$-driven $ASO_4$ formation during high ammonia ($NH_3$) emissions from fertilization. Between 2006 and 2019, mean $\Delta^{17}O(ASO_4)$ was predicted to increase by up to 2‰, driven by declining sulfur dioxide ($SO_2$) emissions from regulatory measures. Model comparisons with historical measurements show reasonable agreement in the acidic southeastern US (RMSE = 0.20‰, Baton Rouge, LA). However, the model overpredicts $\Delta^{17}O(ASO_4)$ in the Western US with RMSE values of 0.36‰ (La Jolla, CA) and 1.9‰ (White Mountain Research Center, CA). This overestimation suggests an excessive model response to aqueous $S(IV) + O_3$ reactions. These findings underscore the diagnostic potential of $\Delta^{17}O(ASO_4)$ for assessing sulfate formation mechanisms and pinpointing shortcomings in chemical transport models. However, $\Delta^{17}O(ASO_4)$ observations across the United States remain exceedingly limited, with most available data dating back to the late 1990s and early 2000s, highlighting the need for renewed measurement efforts.

## 1 Introduction

Atmospheric sulfate ($SO_4^{2-}$) plays a critical role in climate and air quality. As a major component of aerosols, $SO_4^{2-}$ influences aerosol pH, atmospheric chemistry, and precipitation acidity (Calvo et al., 2013; Weber et al., 2016). $SO_4^{2-}$ aerosols ($ASO_4$) significantly contribute to radiative forcing by scattering sunlight and serving as cloud condensation nuclei, which impacts cloud properties and the Earth's radiation balance (Lohmann & Feichter, 1997; Jones et al., 1994; Kaufman & Tanré, 1994).

The anthropogenic influence on the $ASO_4$ budget, primarily from fossil fuel combustion, has been widely documented, contributing to regional and global climate effects (Langner et al., 1992; Smith et al., 2011). The presence of $ASO_4$ alters cloud albedo and lifetime, affecting regional and global climate patterns through indirect radiative forcing (Jones et al., 1994; Haywood & Boucher, 2000). Additionally, the health impacts of $ASO_4$-containing particles underscore their importance in air quality management (Reiss et al., 2007). The formation of $ASO_4$ is influenced by complex interactions with secondary organic

aerosols (SOA) and other atmospheric components. Emerging research highlights the significant role of highly oxygenated organic molecules (HOMs) in enhancing $ASO_4$ formation under humid conditions (Hallquist et al., 2009; Bianchi et al., 2019). These interactions highlight the complex connections between $ASO_4$, atmospheric chemistry, and climate dynamics. Despite a 70% reduction in $ASO_4$ concentrations over the past 15 years, aerosol acidity has remained high, primarily due to the buffering effect of ammonia partitioning between the gas and particle phases (Weber et al., 2016). This persistent acidity

impacts both air quality and health, as it enhances the solubility of harmful metals and promotes acid-catalyzed chemical reactions in the atmosphere.

     Despite their significance, atmospheric chemistry models often face significant challenges in accurately reproducing $ASO_4$ concentrations, potentially due to uncertainties surrounding $ASO_4$ formation mechanisms (Harris et al., 2013; Li et al., 2020;

Vannucci et al., 2024). $ASO_4$ can originate from both primary emissions and secondary formation. Primary sources include natural emissions, such as sea salt, volcanic eruptions, and soil dust (Alexander et al., 2005; Arimoto et al., 2001; Savarino et al., 2003), as well as anthropogenic emissions from fossil fuel combustion (Langner et al., 1992; Smith et al., 2011; Solfen et al., 2011). Secondary $ASO_4$ formation involves complex oxidation processes that occur in both the gas phase and the aqueous phase. In the gas phase, sulfur dioxide ($SO_2$) is oxidized by hydroxyl radicals ($\bullet OH$), producing sulfuric acid ($H_2SO_4$). This

sulfuric acid can either condense to form new particles or add mass to existing aerosols. The rate of this process is highly dependent on environmental conditions such as temperature and pH, which introduces significant uncertainties in predicting $ASO_4$ concentrations (Seigneur & Saxena, 1988). For instance, Vannucci et al. (2024) demonstrated that temperature plays a crucial role in modulating $ASO_4$ aerosol concentrations, particularly during summertime pollution episodes, where aerosol composition and temperature sensitivity can significantly impact model accuracy. Aqueous-phase $ASO_4$ formation occurs

when dissolved sulfur species ($S(IV) = SO_2 \cdot H_2O + HSO_3^- + SO_3^{2-}$) are oxidized by molecules including ozone ($O_3$), hydrogen peroxide ($H_2O_2$), and oxygen, catalyzed by transition metal ions (TMI) (e.g., $Fe^{3+}$ and $Mn^{2+}$). The role of other oxidants, such as hypohalous acids (HOX, X = Cl and Br), is increasingly recognized, particularly in marine boundary layers (Chen et al., 2016; Ishino et al., 2017). Chen et al. (2016) highlighted the significant contribution of HOX in $ASO_4$ formation in the remote marine boundary layer, estimating that 33-50% of $ASO_4$ is produced via this pathway. This suggests that HOX may play a

larger role in $ASO_4$ formation than previously recognized. Additionally, aqueous oxidation of $SO_2$ by nitrogen dioxide ($NO_2$) has also been proposed as a potential pathway, particularly under polluted and low-oxidant wintertime conditions (Sarwar et al., 2013). Although generally less important than $H_2O_2$ and $O_3$ oxidation, this pathway may contribute to $ASO_4$ formation in specific environments and conditions.

Sensitivity analyses have shown that the rate of aqueous-phase $ASO_4$ formation is particularly influenced by pH, oxidant availability, and environmental conditions, further complicating $ASO_4$ modeling (Pandis & Seinfeld, 1989). Harris et al. (2013) showed that TMI-catalyzed oxidation can dominate under specific conditions, particularly in the presence of coarse dust particles, significantly altering $ASO_4$ formation rates in cloud droplets. Similarly, Li et al. (2020) highlighted the critical role of TMI-driven $SO_2$ oxidation during haze episodes, where such pathways can account for up to 50% of $ASO_4$ production under

polluted conditions. Heterogeneous reactions on aerosol surfaces may also play a critical role in $ASO_4$ formation (Harris et al., 2013). These surface reactions involve the interaction of gaseous sulfur species with aerosols, significantly influencing $ASO_4$ formation and elevating the complexity of predicting $ASO_4$ concentrations. Meidan et al. (2019) emphasized the importance of Criegee intermediates (CIs) in $ASO_4$ formation, particularly in nocturnal power plant plumes, where $SO_2$ is oxidized under conditions with minimal photochemical activity. This study revealed that CIs could account for a significant portion of $ASO_4$

production in the absence of sunlight. Additionally, Liu et al. (2019) examined the role of stabilized Criegee intermediates (sCIs) in $ASO_4$ formation in the Beijing-Tianjin-Hebei region, showing that under certain atmospheric conditions, sCI-driven $SO_2$ oxidation can contribute substantially to secondary $ASO_4$ production, adding another layer of complexity to $ASO_4$ formation models. These interactions highlight the challenges in modeling $ASO_4$ concentrations, as heterogeneous reactions, TMIs, and Criegee intermediates all contribute to the uncertainty in atmospheric $ASO_4$ predictions.


The use of oxygen isotope mass-independent fractionation ($\Delta^{17}O = \delta^{17}O - 0.52 \times \delta^{18}O$) has emerged as a promising tool to explore atmospheric $ASO_4$ formation pathways (Alexander et al., 2004; Barkan & Luz, 2003; Kaiser et al., 2004; Michalski et al., 2003; Morin et al., 2007; Savarino et al., 2007; Walters et al., 2019; Weston, 2006). This isotopic indicator is crucial for tracking $ASO_4$ formation, providing a refined tool for model evaluation and prediction. This is because $\Delta^{17}O$ has distinct values

associated with different oxidation processes, making it a powerful tool in understanding $ASO_4$ production mechanisms. The dominant source of $\Delta^{17}O$ in the lower atmosphere derives from $O_3$ formation. The average $\Delta^{17}O(O_3)$ near the surface is approximately 26‰ (Vicars & Savarino, 2014). This contrasts with other tropospheric oxidants, which have $\Delta^{17}O$ values near 0‰. Hydrogen peroxide ($H_2O_2$) has a $\Delta^{17}O$ value of about 1.6‰ due to the influence of $O_3$ involved in $H_2O_2$ formation (Savarino & Thiemens, 1999). Laboratory studies have shown that oxidants will proportionally transfer their $\Delta^{17}O$ values into

the $ASO_4$ product. Table 1 summarizes the $\Delta^{17}O$ ranges associated with major tropospheric $ASO_4$ production pathways based on oxygen isotopic mass balance (Alexander et al., 2005; Alexander et al., 2009; Ishino et al., 2017; Savarino et al., 2000; Walters et al., 2019). The gas-phase oxidation of $SO_2$ by OH and metal-catalyzed $O_2$ oxidation yields $\Delta^{17}O(ASO_4)$ values near 0‰, indicating a negligible transfer of the $\Delta^{17}O$ signature. Similarly, aqueous-phase oxidation of $SO_2$ by hypohalous acids (HOX) results in $\Delta^{17}O(ASO_4)$ values around 0‰. In contrast, aqueous-phase oxidation involving $H_2O_2$ and $O_3$ exhibits

significantly higher $\Delta^{17}O$ values. $H_2O_2$ oxidation produces $\Delta^{17}O(ASO_4)$ values around 0.8‰, while $O_3$ oxidation results in $\Delta^{17}O(ASO_4)$ values of about 6.5‰. These distinctions enable the ability to track $ASO_4$ formation.

Previous studies have utilized $\Delta^{17}O(ASO_4)$ observations to evaluate the impact of anthropogenic emissions on $ASO_4$ production routes. In polluted regions, anthropogenic emissions of metals such as $Fe^{3+}$ and $Mn^{2+}$ enhance $O_2$-catalyzed $ASO_4$ formation, particularly in the Northern Hemisphere during winter. This metal-catalyzed $ASO_4$ formation can suppress $ASO_4$ production via $O_3$ and $H_2O_2$ pathways, impacting $\Delta^{17}O(ASO_4)$ values and complicating model predictions (Savarino et al., 2000). Furthermore, ship emissions, which have been underrepresented in atmospheric models, significantly contribute to $ASO_4$ source in marine environments. Triple-oxygen isotope measurements suggest these emissions play a larger role in $ASO_4$ production than previously recognized, with implications for air quality and climate modeling (Dominguez et al., 2008). To fully utilize the diagnostic potential of $\Delta^{17}O(ASO_4)$, a comprehensive model framework is essential for interpreting $ASO_4$ formation. Previous models, such as GEOS-Chem, have incorporated $\Delta^{17}O$ tracking to investigate $ASO_4$ formation pathways, highlighting the growing importance of metal-catalyzed $O_2$ oxidation in polluted regions, which surpasses the traditional $O_3$ and $H_2O_2$ pathways (Sofen et al., 2011). Despite rising tropospheric $O_3$ levels since preindustrial times, $\Delta^{17}O(ASO_4)$ values in the Arctic have declined due to enhanced metal-catalyzed $ASO_4$ formation. Recent studies have applied $\Delta^{17}O$ of $ASO_4$ in chemical transport models to explore long-term changes and regional processes, including GEOS-Chem simulations coupled with ice core observations (Hattori et al., 2021; Peng et al., 2023) and CMAQ applications in East Asia (Itahashi et al., 2022; Lin et al., 2025). These works highlight the diagnostic potential of $\Delta^{17}O$ across diverse regions and timescales. Building upon these advances, our study presents the first CMAQ simulations of $\Delta^{17}O(ASO_4)$ within the contiguous United States over full annual cycles for 2006 and 2019, allowing for the assessment of seasonal and spatial patterns of $ASO_4$ oxidation pathways in response to emission reductions.

In this work, $\Delta^{17}O$ tracking has now been incorporated into the Community Multiscale Air Quality Model (CMAQ), a 3-D atmospheric chemistry transport model. CMAQ offers high spatial and temporal resolution, which is critical for studying $ASO_4$ formation pathways and for validating model predictions through observational testing (Appel et al., 2021). This study aims to refine $ASO_4$ formation modeling by integrating $\Delta^{17}O$ tracking into CMAQ, thus improving predictions of $ASO_4$ dynamics and reducing uncertainties in atmospheric chemistry models with a focus on the contiguous US. The spatiotemporal $\Delta^{17}O$ values predicted by CMAQ will help validate model predictions and advance our understanding of atmospheric $ASO_4$ chemistry and its connection to air quality and deposition. Establishing reference $\Delta^{17}O$ values across the contiguous United States (CONUS) is a key outcome of this study, as it lays the groundwork for future research, enhances air quality and deposition-related studies, and contributes to improved air quality management strategies by providing a more accurate representation of $ASO_4$ formation across different regions.

**Table 1: Major $ASO_4$ formation pathways and their associated $\Delta^{17}O$ signatures. The pathways that are included in the CMAQ model using the cb6r5-ae7-aq mechanism are indicated.**

| Pathway | Reaction | $\Delta^{17}O$ (‰) | Notes | Included in CMAQ (cb6r5-ae7) |
|---|---|---|---|---|

| Phase | Reaction | $\Delta^{17}O$ | Notes | |
|---|---|---|---|---|
| Gas-Phase | $SO_2 + \cdot OH \rightarrow SO_4^{2-} + HO_2$ | ~0 | Dominant in photochemically active regions; negligible $\Delta^{17}O$ signature | Yes |
| Gas-Phase | $SO_2 + sCI$ (Stabilized Criegee Intermediates) $\rightarrow SO_4^{2-}$ | ~0 | Can enhance secondary $ASO_4$ formation in regions influenced by biogenic VOC emissions; negligible $\Delta^{17}O$ signature | NO |
| Aqueous-Phase | $HSO_3^- + H_2O_2 \rightarrow SO_4^{2-} + H_2O$ | 0.8 | Lower $\Delta^{17}O$ value, dominant under humid/cloudy conditions. | Yes |
| Aqueous-Phase | $SO_3^{2-} + O_3 \rightarrow SO_4^{2-} + O_2$ | 6.5 | Higher $\Delta^{17}O$ value, significant for cloud chemistry. | Yes |
| Aqueous-Phase | $SO_3^{2-} + O_2$ (TMI = Transition Metal Ions, e.g., $Fe^{3+}$ and $Mn^{2+}$) $\rightarrow SO_4^{2-}$ | ~0 | Important in metal-rich aerosols; negligible $\Delta^{17}O$ signature | Yes |
| Aqueous-Phase | $SO_3^{2-} + NO_2 \rightarrow SO_4^2 + NO$ | ~0 | Relevant under polluted and low-oxidant wintertime conditions; negligible $\Delta^{17}O$ signature | No |
| Aqueous-Phase | $SO_3^{2-} + HOX$ (X = Br, Cl) $\rightarrow SO_4^{2-}$ | ~0 | Dominant in marine environments with halogen chemistry; negligible $\Delta^{17}O$ signature | No |
| Heterogeneous | $SO_2$ (surface) + Organic peroxides $\rightarrow SO_4^{2-}$ | ~0 | Significant role in submicron aerosol $ASO_4$ formation; negligible $\Delta^{17}O$ signature. | No |
| Heterogeneous | $SO_2$ (surface) + $H_2O_2$ or $O_3$ on aerosols $\rightarrow SO_4^{2-}$ | 0.8 - 6.5 | Highly variable; depends on aerosol composition and environmental conditions. | No |

130

## 2 Methods

### 2.1 Model Description and EQUATES 2019 Dataset

This study utilizes the CMAQ (Community Multiscale Air Quality) version 5.4 model to simulate $ASO_4$ formation and its $\Delta^{17}O$ values across the contiguous United States (CONUS). The CMAQ model is configured with the cb6r5_ae7_aq chemical mechanism, which stands for Carbon Bond 6 revision 5, with aerosol 7 for standard cloud chemistry (Yarwood et al., 2010). This mechanism encompasses both gas-phase and aqueous-phase oxidation processes of $SO_2$, which are essential for accurately modeling $ASO_4$ formation. Specifically, it involves the oxidation of $SO_2$ by •OH in the gas phase and by $H_2O_2$ and $O_3$ in cloud droplets and aqueous environments. Cloud water pH in CMAQ is calculated dynamically within the default cloud chemistry module, which is based on the work of Walcek and Taylor (1986) and assumes instantaneous equilibrium among gas, aqueous, and ionic species. The pH is determined by the charge balance between dissolved acidic and basic ions. As S(IV) is oxidized to S(VI) and additional species are scavenged from interstitial aerosols, the pH evolves dynamically throughout cloud processing. The resulting pH fields respond to emissions and meteorological variability, directly governing the relative importance of the $H_2O_2$ and $O_3$ oxidation pathways for aqueous S(IV) oxidation. Previous evaluations have demonstrated that CMAQ accurately reproduces observed cloud droplet acidity, with differences generally within 0.5 pH units across multiple sites in the United States (Pye et al., 2020). The model's ability to capture these complex interactions facilitates a detailed assessment of $ASO_4$ dynamics under various atmospheric conditions (Appel et al., 2021).

The CMAQ simulations are based on the EQUATES (EPA's Air Quality Time Series Project) dataset, which provides a comprehensive and high-resolution emissions inventory derived from the 2017 National Emissions Inventory (NEI) (Benish et al., 2022; Foley et al., 2023). This dataset spans over two decades and provides detailed information on both natural and anthropogenic emissions, including those from industrial sources, vehicular traffic, power plants, and wildfires. It also accounts for seasonal and regional variations in emissions, enhancing the model's accuracy. The EQUATES 2019 dataset supplies critical inputs for CMAQ simulations, including emissions data, meteorological variables, as well as boundary and initial conditions, capturing pollutant variability across different seasons and regions.

Meteorological inputs for the CMAQ simulations were integrated from the Weather Research and Forecasting (WRF) model version 4.1.1. This integration provides detailed representations of temperature, wind speed, relative humidity, cloud cover, and precipitation rates. These meteorological factors influence cloud formation, pollutant dispersion, and oxidation processes. Boundary and initial conditions for the CMAQ model were sourced from EQUATES to ensure accurate representation of pollutant inflows and outflows at the edges of the modeling domain. The initial conditions were established through a spin-up period starting on December 15, 2018, providing accurate starting concentrations for the 2019 simulation period. The CMAQ simulations were conducted at a resolution of 12 × 12 km over the CONUS domain using the Hyperion high-performance

computing cluster at the University of South Carolina. This advanced computing infrastructure enabled the processing of large datasets and the execution of complex simulations necessary for this study.

## 2.2 Implementation of the Sulfur Tracking Mechanism (STM)

The Sulfur Tracking Mechanism (STM), utilized in the CMAQ model, provides a detailed analysis of $ASO_4$ formation pathways in the atmosphere (Appel et al., 2021). It distinguishes between various aqueous-phase and gas-phase formation processes and assesses contributions from emissions, initial conditions, and boundary conditions, offering valuable insights into the roles of these factors in overall $ASO_4$ production (Table 2). The sulfur budget comprises 14 $ASO_4$ species (AE) and 1 nonreactive $ASO_4$ species (NR), as documented in the CMAQ repository (https://github.com/USEPA/CMAQ/blob/main/CCTM/src/MECHS/README.md). The STM output includes hourly simulations of the 15 tagged $ASO_4$ species across the model domain, which were then aggregated into monthly averages to analyze spatial and temporal variations in $ASO_4$ production. STM allows for an efficient way for the model to distinguish the contributions of different chemical pathways and emission contributions to $ASO_4$. This approach also enables a seamless calculation of $\Delta^{17}O$ of $ASO_4$.

A known bookkeeping bug in the STM implementation in CMAQ v5.4 resulted in the systematic underestimation of $ASO_4$ formed via the gas-phase $SO_2 + OH$ pathway, despite the pathway being chemically active in the model. This issue has been documented by the CMAQ development team (https://github.com/USEPA/CMAQ/wiki/CMAQ-Release-Notes:-Process-Analysis-&-Sulfur-Tracking-Model-(STM)) and has since been corrected in version 5.5. In our study, we fixed the issue by adjusting the call order of STM update routines in the sciproc.F module (matching the v5.5 fix) and ran the simulations using the corrected code. The updated STM diagnostic module used in this work is publicly available for reproducibility at: https://doi.org/10.5281/zenodo.14954960.

**Table 2: Overview of the $ASO_4$ species in the Sulfur Tracking Mechanism (STM) incorporated into CMAQ.**

| Name | Group | Mode | Pathway |
|------|-------|------|---------|
| ASO4GASI | AE | Aitken | condensation of gas-phase reaction with OH |
| ASO4EMISI | AE | Aitken | source emission |
| ASO4ICBCI | AE | Aitken | initial conditions and boundary conditions |
| ASO4AQH2O2J | AE | Accumulation | $H_2O_2$ |
| ASO4AQO3J | AE | Accumulation | $O_3$ |
| ASO4AQFEMNJ | AE | Accumulation | $O_2$ catalyzed by $Fe^{3+}$ and $Mn^{2+}$ |
| ASO4AQMHPJ | AE | Accumulation | methyl hydrogen peroxide (MHP) |
| ASO4AQPAAJ | AE | Accumulation | peroxyacetic acid (PAA) |

| ASO4GASJ | AE | Accumulation | condensation of gas-phase reaction with OH |
|---|---|---|---|
| ASO4EMISJ | AE | Accumulation | source emission |
| ASO4ICBCJ | AE | Accumulation | initial conditions and boundary conditions |
| ASO4GASK | AE | Coarse | condensation of gas-phase reaction with OH |
| ASO4EMISK | AE | Coarse | source emission |
| ASO4ICBCK | AE | Coarse | initial conditions and boundary conditions |
| SULF_ICBC | NR | N/A | sulfuric acid vapor (SULF) from initial conditions and boundary conditions |

## 2.3 Calculation and Analysis of $\Delta^{17}O(ASO_4)$

The fractional contributions of each pathway, obtained from the STM, are used to calculate $\Delta^{17}O(ASO_4)$ across different grid cells.

$$f_i(lat, lon, height, time) = \frac{X_i}{\sum_{i=1}^{n} X_i}$$

(1)

where $f_i$ represents the fractional contribution of pathway $i$; $X_i$ is the amount of $ASO_4$ produced by pathway $i$; and $\sum_{i=1}^{n} X_i$ is the total Aitken mode and accumulation mode $ASO_4$ produced by all pathways except initial conditions and boundary conditions in each grid cell.

The gas-phase oxidation of $SO_2$ by •OH radical results in $ASO_4$ with no significant $\Delta^{17}O$ enrichment (~0‰). $ASO_4$ formed via aqueous-phase oxidation by $O_3$ has a $\Delta^{17}O$ value of ~6.5‰, indicating significant cloud chemistry processes. $ASO_4$ formed via aqueous-phase oxidation by $H_2O_2$ has a $\Delta^{17}O$ value of ~0.8‰. Metal-catalyzed oxidation of $SO_2$ by $O_2$ in metal-rich environments results in a $\Delta17O$ value of ~0‰ and does not exhibit a transfer of mass-independent fractionation signature.

Although previous studies reported slightly negative $\Delta^{17}O$ values (-0.1‰; Hattori et al., 2021; Itahashi et al., 2022), this pathway contributes less than 10% to total $ASO_4$ formation in our simulations, leading to a negligible (<0.01‰) effect on the modeled $\Delta^{17}O(ASO_4)$. Therefore, it is approximated as 0‰ in this study. Heterogeneous reactions, such as those involving organic peroxides on aerosol surfaces, contribute to $ASO_4$ formation and are expected to have a $\Delta^{17}O$ ~0‰. Although the fractional contributions ($f_i$) include all $ASO_4$ formation pathways diagnosed by the Sulfur Tracking Mechanism (STM), only

$H_2O_2$ and $O_3$ carry non-zero $\Delta^{17}O$ signatures, all other pathways are assigned $\Delta^{17}O \approx 0$‰. Therefore, the $\Delta^{17}O(ASO_4)$ is calculated using the following equation:

$$\Delta^{17}O(ASO_4) = f_{ASO4AQH2O2J} \times 0.8‰ + f_{ASO4AQO3J} \times 6.5‰$$

(2)

where ASO4AQH2O2J represents ASO$_4$ formed through the oxidation of SO$_2$ by H$_2$O$_2$; ASO4AQO3J represents ASO$_4$ formed through oxidation by O$_3$; the constants 0.8‰ and 6.5‰ correspond to the characteristic $\Delta^{17}$O values for each pathway.

## 3 Results and Discussion

### 3.1 Predicted Fractional ASO$_4$ Formation and $\Delta^{17}$O(ASO$_4$) in the Contiguous US in 2019

ASO$_4$ production in the contiguous United States arises from a combination of primary emissions and secondary formation pathways, the latter being dominated by H$_2$O$_2$- and O$_3$-driven aqueous S(IV) oxidation and gas-phase oxidation of SO$_2$ via •OH (Fig. 1). These secondary reactions occur within cloud water, where SO$_2$ is oxidized by H$_2$O$_2$, O$_3$, and by O$_2$ (catalyzed by TMI). Compared to these dominant pathways, the TMI-catalyzed oxidation and reactions involving organic peroxides, such as methyl hydrogen peroxide (MHP) and peroxyacetic acid (PAA), have a minimal impact on ASO$_4$ production. While primary emissions contribute little overall, they exhibit localized hotspots in certain regions.

The fractional contributions of ASO$_4$ formation pathways demonstrate distinct spatial patterns that align with the predicted $\Delta^{17}$O(ASO$_4$) variability. The H$_2$O$_2$ pathway ($f_{S(IV)+H2O2}$) is the most dominant, accounting for 35.4±14.0% of the ASO$_4$ formation across the domain (Fig. 1). This pathway is particularly influential in the Gulf Coast States, where abundant cloud cover and water vapor, acidic conditions (cloud pH < 6), and high concentrations of H$_2$O$_2$ (Fig. S1, Fig. S2) support the oxidation of S(IV) in cloud droplets. The highest $f_{S(IV)+H2O2}$ in these regions contribute to the low $\Delta^{17}$O(ASO$_4$) values below 1‰, due to the relatively lighter $\Delta^{17}$O(ASO$_4$) resulted from the H$_2$O$_2$ pathway (0.8‰). Gas-phase oxidation of SO$_2$ by •OH ($f_{SO2+OH}$) contributes to 34.4%±9.5 of the ASO$_4$ production across the domain (Fig. 1), exhibiting clear seasonal variability under photochemically active conditions, with the highest contributions occurring in summer (up to ~75 %) and lowest in winter (< 25 %) (Fig. 5). The O$_3$ pathway ($f_{S(IV)+O3}$) is the third most significant, contributing approximately 18.7±5.2% to the ASO$_4$ formation across the domain (Fig. 1). The highest $f_{S(IV)+O3}$ occurs in the Western States, due to the high O$_3$ concentration and high cloud pH (Fig. S1), which facilitates the aqueous oxidation of S(IV) by O$_3$. With a higher $\Delta^{17}$O(ASO$_4$) value resulting from the S(IV) + O$_3$ pathway of 6.5‰, the higher $f_{S(IV)+O3}$ in these regions results in elevated $\Delta^{17}$O(ASO$_4$) values, typically above 2‰. Minor pathways, such as those involving TMI, MHP, and PAA, contribute 2.3±1.8%, 0.25±0.25%, and 0.17±0.12%, respectively (Fig. 1), to ASO$_4$ formation across the US continuous domain. Primary ASO$_4$ emissions account for 8.7±6.4% of total ASO$_4$ (Fig. 1), with substantial contributions originating from urban and industrial regions. High SO$_2$ emissions from anthropogenic activities in these areas elevate the role of primary ASO$_4$, and their impact on $\Delta^{17}$O(ASO$_4$) is correspondingly notable but limited to these localized hotspots.

Cloud pH is a critical determinant of ASO$_4$ formation pathways and $\Delta^{17}$O(ASO$_4$) values, with lower cloud pH favoring the H$_2$O$_2$ pathway and higher cloud pH supporting the O$_3$ pathway (Seigneur & Saxena, 1988; Fahey & Pandis, 2001). The

concentration of ASO$_4$ plays a dominant role in lowering cloud pH, primarily due to its origin from sulfuric acid (H$_2$SO$_4$). As a strong acid, H$_2$SO$_4$ dissociates completely, releasing significant amounts of hydrogen ions (H$^+$) and causing substantial acidification of cloud water. In regions such as the Northeast, Southeast, and Midwest, relatively high SO$_2$ emissions result in elevated ASO$_4$ concentrations, which further favor the dominance of the H$_2$O$_2$ oxidation pathway over O$_3$, thereby sustaining low $\Delta^{17}$O(ASO$_4$) values in the Northeast and Southeast (Fig. S1). This is due to the efficient conversion of dissolved S(IV)

species to ASO$_4$, primarily through the aqueous S(IV)+H$_2$O$_2$ pathway under acidic cloud water. Frequent cloud occurrence and abundant oxidant availability accelerate SO$_2$ to ASO$_4$ production. These high ASO$_4$ levels contribute significantly to lowering cloud pH in these areas, creating an acidic environment (Fig. S1). In contrast, in the Western States, SO$_2$ emissions and ASO$_4$ concentrations are comparatively lower (Fig. S1). This results in reduced acidification and a higher cloud pH, as the influence of ASO$_4$ on the acidity of cloud water is diminished. Ammonium in cloud water (ANH$_4$), on the other hand, primarily

acts as a buffering agent, mitigating the acidity caused by ASO$_4$ (Fig. S1). NH$_3$ reacts with H$_2$SO$_4$ to form (NH$_4$)$_2$SO$_4$ (ammonium sulfate), which reduces the availability of free H$^+$ and partially neutralizes the acidification caused by ASO$_4$. However, the neutralizing capacity of ANH$_4$ is limited. In regions with high ASO$_4$ concentrations, such as the Northeast and Southeast, the buffering effect of ANH$_4$ is insufficient to fully counteract the strong acidity introduced by ASO$_4$. In the Midwest, where NH$_3$ emissions from agricultural activities, particularly fertilization, are significant, the resulting high

concentrations of ANH$_4$ partially neutralize the acidity from ASO$_4$ (Fig. S1). This interaction raises cloud pH slightly, preventing extreme acidification (Fig. S1). Nevertheless, even in regions with abundant NH$_3$ emissions, cloud water pH typically remains acidic because of the dominant influence of ASO$_4$ and other atmospheric acids.

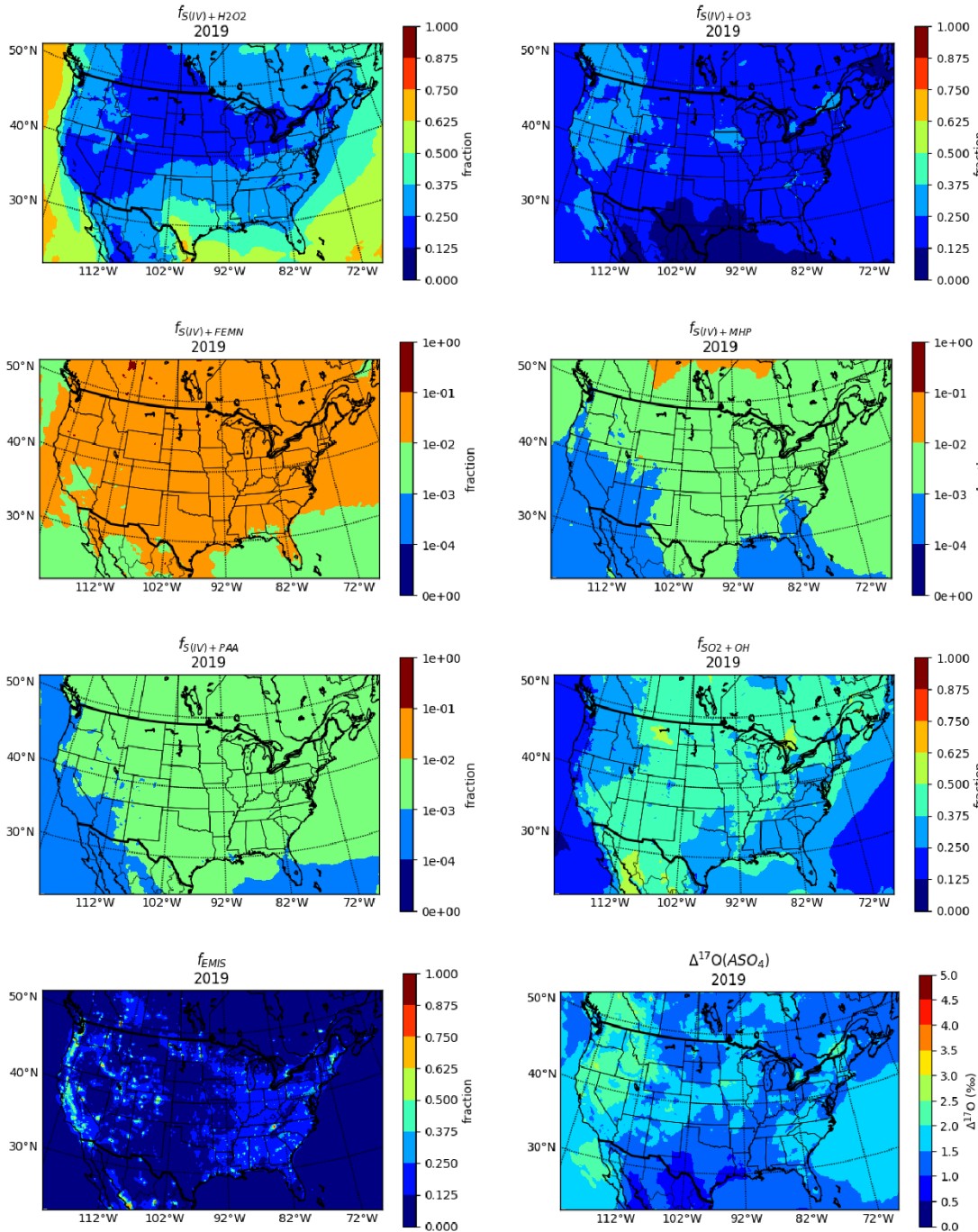

**Fig. 1: The annual fractional contribution from each ASO₄ formation pathway, along with Δ¹⁷O(ASO₄) across the contiguous US in the year 2019, based on CMAQ simulation.**

## 3.2 Seasonal Variation in Fractional ASO$_4$ Formation and $\Delta^{17}$O(ASO$_4$) in the Contiguous US in 2019

ASO$_4$ formation across the contiguous United States exhibits distinct seasonal patterns, shaped by varying contributions from the H$_2$O$_2$ and O$_3$ pathways, as well as shifts in cloud pH and precursor concentrations (Fig. S1). The isotopic composition of ASO$_4$, represented by $\Delta^{17}$O(ASO$_4$), reflects the dominance of specific pathways under different meteorological and chemical conditions. In regions with low cloud pH, the H$_2$O$_2$ pathway dominates, resulting in low $\Delta^{17}$O values (Fig. S1, Fig. 2). Conversely, areas with high cloud pH favor the O$_3$ pathway, resulting in high $\Delta^{17}$O values (Fig. S1, Fig. 2). Meanwhile, gas-phase SO$_2$ + •OH oxidation (Fig. 5) acts as a consistent background process throughout the year. Its near-zero $\Delta^{17}$O signature moderates the contrast between H$_2$O$_2$- and O$_3$-dominated regimes, with a stronger influence during summer when photochemical activity peaks. Together with seasonal variations in cloud pH, ASO$_4$, and ANH$_4$ (Fig. S3, Fig. S4, Fig. S6), this process shapes the overall spatiotemporal pattern of $\Delta^{17}$O(ASO$_4$), reflecting the coupled effects of emissions, atmospheric chemistry, and meteorology.

In January, the Western States exhibit the highest $\Delta^{17}$O(ASO$_4$) values, exceeding 2‰ (Fig. 2). This is driven by the increased importance of the S(IV) + O$_3$ oxidation pathway (Fig. 3), supported by elevated cloud pH levels resulting from low ASO$_4$ concentrations (Fig. S3, Fig. S4). Conversely, the Gulf Coast States show the lowest $\Delta^{17}$O(ASO$_4$) values, typically below 1‰ (Fig. S1), primarily due to the prevalence of the H$_2$O$_2$ pathway (Fig. 4). This pathway dominates under low cloud pH conditions caused by high ASO$_4$ concentrations and limited ANH$_4$ levels (Fig. S3, Fig. S4, Fig. S6). In the Midwest, moderate $\Delta^{17}$O(ASO$_4$) values are shown, reflecting a balance between pathways. Elevated ANH$_4$ levels partially neutralize the acidity from high ASO$_4$ concentrations (Fig. S4, Fig. S6). This neutralization raises cloud pH (Fig. S3), slightly lowering the fractional contribution of the S(IV) + H$_2$O$_2$ pathway (Fig. 4) and contributing to the intermediate $\Delta^{17}$O(ASO$_4$) values. During this period, the SO$_2$ + •OH pathway remains weak due to limited photochemical activity but provides a modest background effect that slightly reduces the isotopic contrast between H$_2$O$_2$- and O$_3$-dominated regimes.

In April, $\Delta^{17}$O values increase significantly, particularly in the Western States, rising above 3‰ (Fig. 2). This trend indicates an enhanced influence of the O$_3$ pathway, supported by elevated cloud pH and increased O$_3$ levels (Fig. 3, Fig. S3, Fig. S8). In contrast, the Gulf Coast States continue to exhibit low $\Delta^{17}$O values (< 1.5‰) (Fig. 2), as the H$_2$O$_2$ pathway remains dominant due to persistently low cloud pH (Fig. 4, Fig. S3). This acidity is driven by high ASO$_4$ concentrations and low ANH$_4$ concentrations (Fig. S4, Fig. S6). Meanwhile, in the Midwest, cloud pH begins to rise as increased NH$_3$ levels, partially neutralizing the acidity from ASO$_4$ and shifting the balance of ASO$_4$ formation pathways (Fig. S3, Fig. S4, Fig. S7). The influence of the SO$_2$ + •OH pathway decreases relative to January (Fig. 5), as O$_3$ oxidation becomes more dominant, thereby exerting a weaker moderating effect on the isotopic contrast across regions.

In July, $\Delta^{17}O$ values decrease in the Western States as the $f_{S(IV)+H2O2}$ increases compared to April (Fig. 2). This shift is driven by higher water vapor levels and increased cloud cover (Fig. S10), despite the regional consistently high cloud pH (Fig. S3). In the Gulf Coast States, $\Delta^{17}O$ values remain low, below 1.5‰ (Fig. S1), highlighting the continued dominance of the $H_2O_2$ pathway (Fig. 4) under conditions of abundant water vapor (Fig. S10), frequent cloud cover (Fig. S11), and persistently low cloud pH (Fig. S3). In the Midwest, cloud pH continues to rise from April (Fig. S3), driven by increasing $NH_3$ concentrations (Fig. S7), which partially neutralize the acidity caused by $ASO_4$ (Fig. S4). This elevation in cloud pH enhances the activity of the $O_3$ pathway (Fig. S2), leading to an increase in $\Delta^{17}O$ values compared to April. At the same time, the $SO_2 + \bullet OH$ pathway reaches its maximum importance (Fig. 5) under strong photochemical conditions, offsetting the isotopic enrichment from $O_3$ oxidation and contributing to the relative decrease in $\Delta^{17}O(ASO_4)$ compared to April.

In October, $\Delta^{17}O$ values in the Western States increase compared to July but remain slightly lower than in April (Fig. 2). This change is attributed to the enhanced $f_{S(IV)+O3}$ (Fig. S2), supported by high cloud pH and low $ASO_4$ concentrations (Fig. S3, Fig. S4). In the Gulf Coast States, $\Delta^{17}O$ values remain low (Fig. 2), reflecting the continued dominance of the $H_2O_2$ pathway under acidic conditions sustained by high $ASO_4$ levels and low $ANH_4$ concentrations (Fig. S3, Fig. S4, Fig. S6). In the Midwest, decreasing $NH_3$ levels from July reduce the neutralization of acidity (Fig. S3, Fig. S7), making conditions less favorable for $O_3$-driven $ASO_4$ formation (Fig. 3). This results in lower cloud pH (Fig. S3) and diminished $\Delta^{17}O$ values compared to earlier months. As photochemical activity weakens, the relative contribution of the $SO_2 + \bullet OH$ pathway declines (Fig. 5), reducing its moderating effect and allowing $O_3$-driven isotopic enrichment to strengthen in high-pH regions.

Seasonal variations in $ASO_4$ formation and $\Delta^{17}O(ASO_4)$ highlight the interplay of chemical drivers and meteorological conditions. The dominance of the $H_2O_2$ pathway in acidic, $ASO_4$-rich regions, such as the Gulf Coast States, leads to low $\Delta^{17}O$ values year-round. In contrast, the $O_3$ pathway prevails in higher pH regions such as the Western States, driving elevated $\Delta^{17}O$ values, particularly in April. The Midwest experiences transitional conditions, where cloud pH and $NH_3$ concentrations modulate the relative contributions of $ASO_4$ formation pathways. Alongside these seasonal and spatial contrasts, the gas-phase $SO_2 + \bullet OH$ pathway acts as a persistent, near-zero-$\Delta^{17}O$ background that offsets isotopic enrichment from $O_3$ oxidation, particularly during summer when photochemical activity peaks. These findings underscore the dynamic nature of $ASO_4$ chemistry across seasons, emphasizing the importance of emissions, atmospheric composition, and cloud chemistry in shaping regional and seasonal patterns of $ASO_4$ formation.

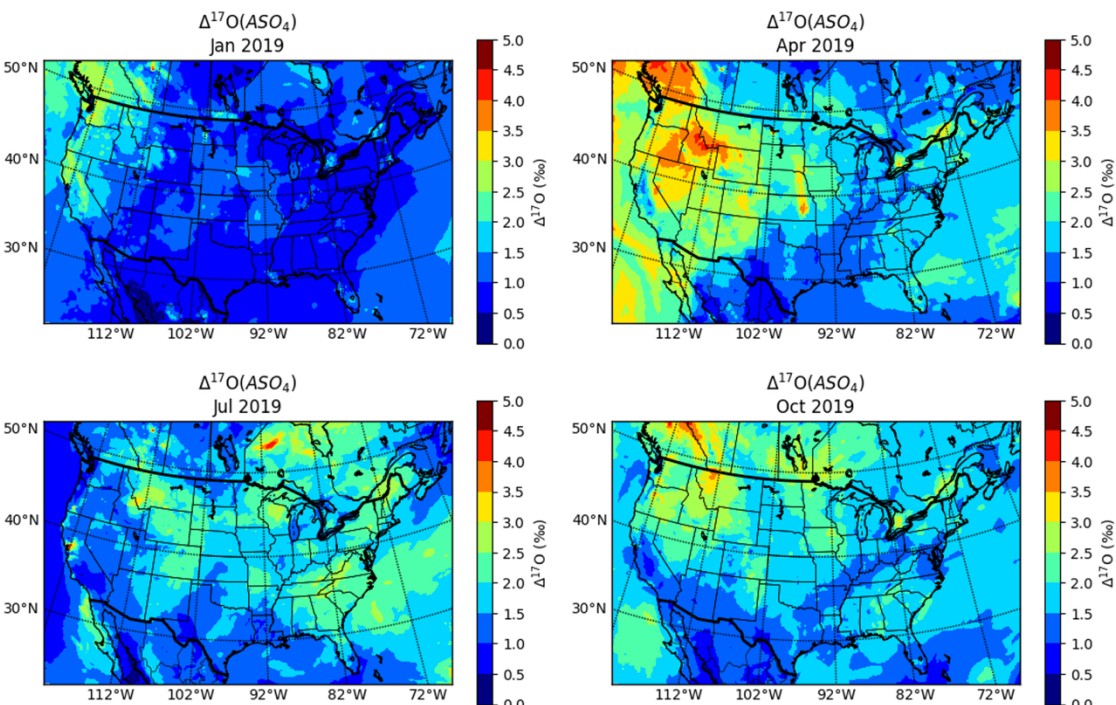

Fig. 2: The simulated $\Delta^{17}O(ASO_4)$ values across the contiguous US for the year 2019 in each season (winter: Jan, spring: Apr, summer: July, fall: Oct), based on CMAQ simulation.

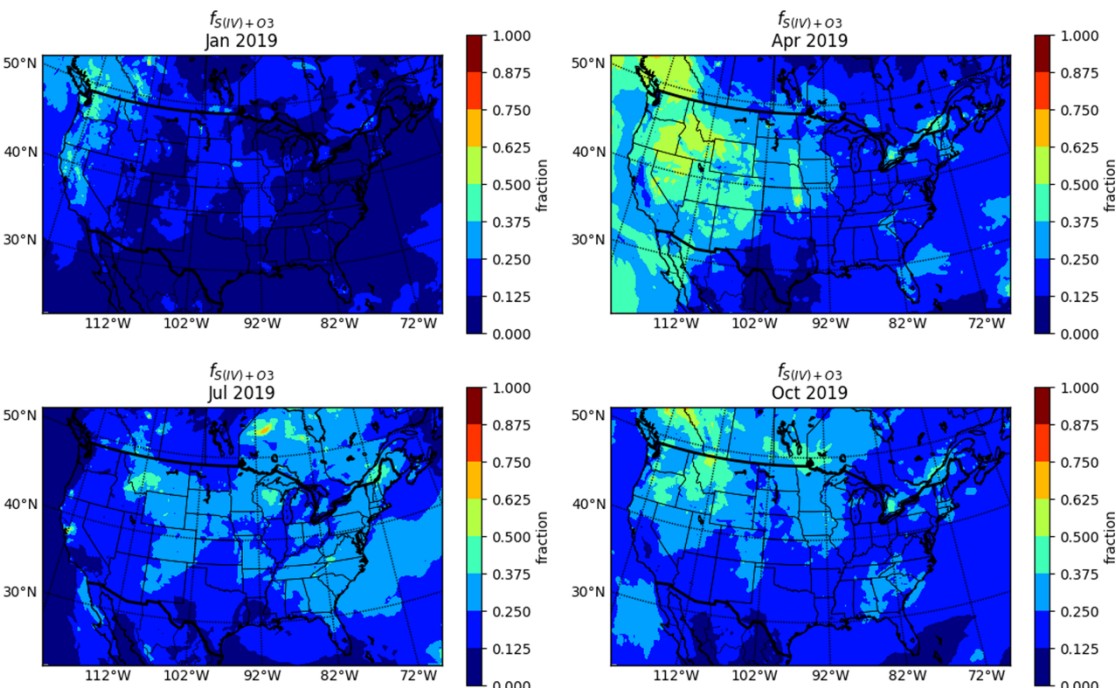

**Fig. 3: The fraction of ASO₄ formation from S(IV)+O₃ pathway across the contiguous US for the year 2019 in each season (winter: Jan, spring: Apr, summer: July, fall: Oct), based on CMAQ simulation.**

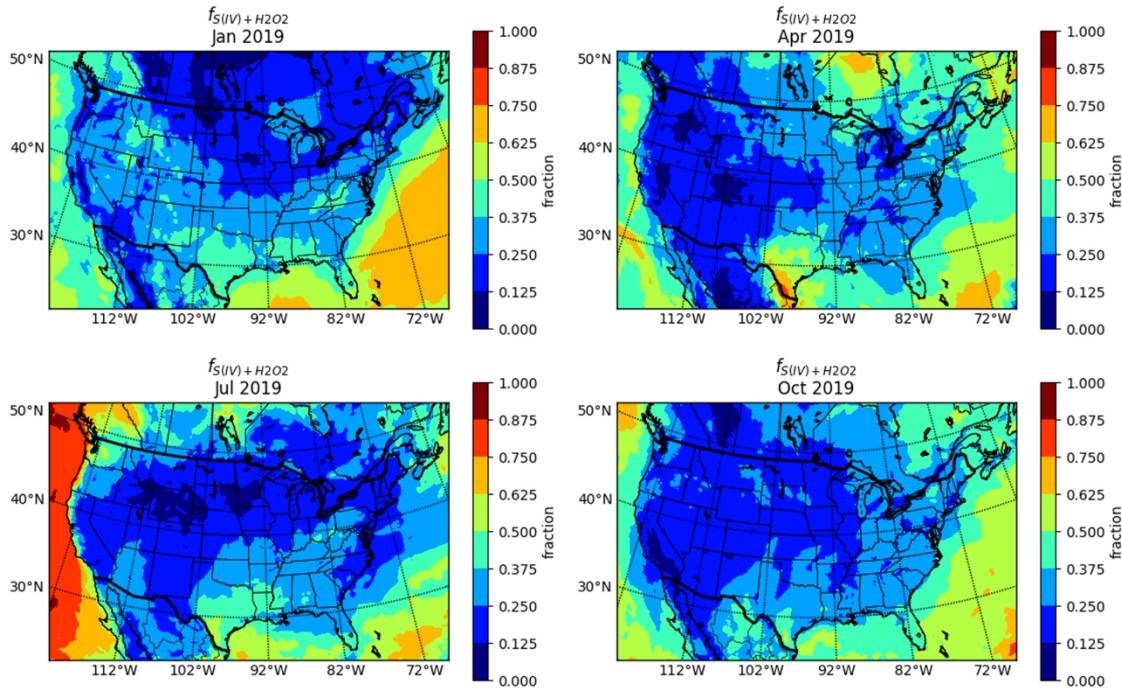

**Fig. 4: The fraction of ASO₄ formation from S(IV)+H₂O₂ pathway across the contiguous US for the year 2019 in each season (winter: Jan, spring: Apr, summer: July, fall: Oct), based on CMAQ simulation.**

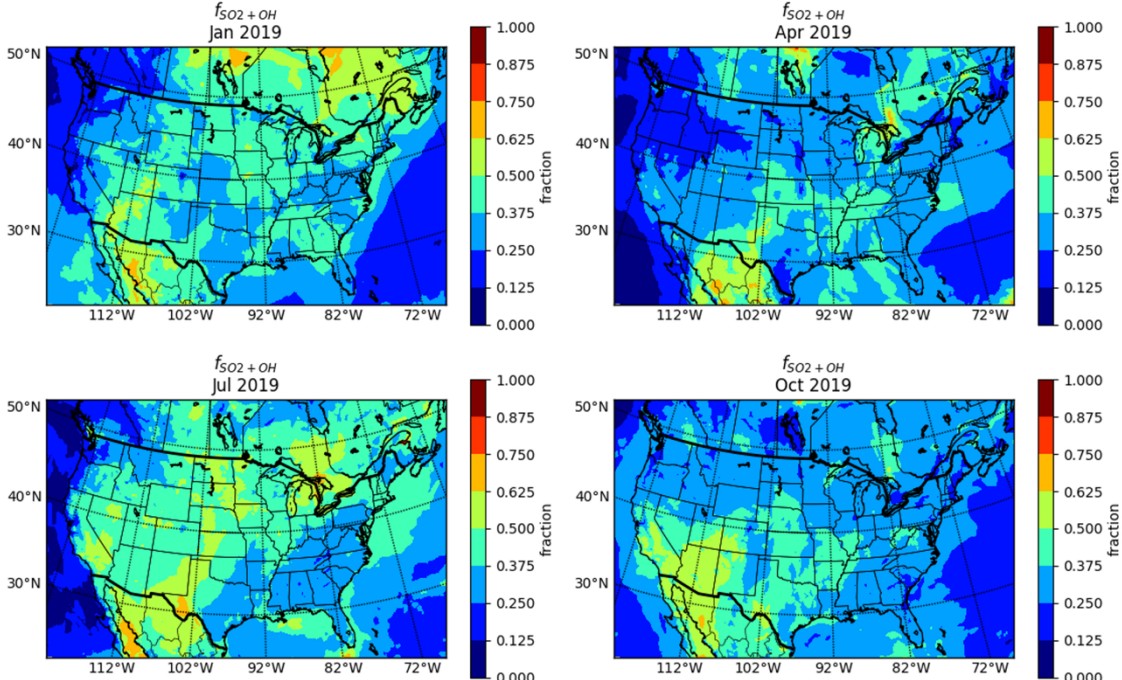

**Fig. 5: The fraction of ASO₄ formation from SO₂ + OH pathway across the contiguous US for the year 2019 in each season (winter: Jan, spring: Apr, summer: July, fall: Oct), based on CMAQ simulation.**

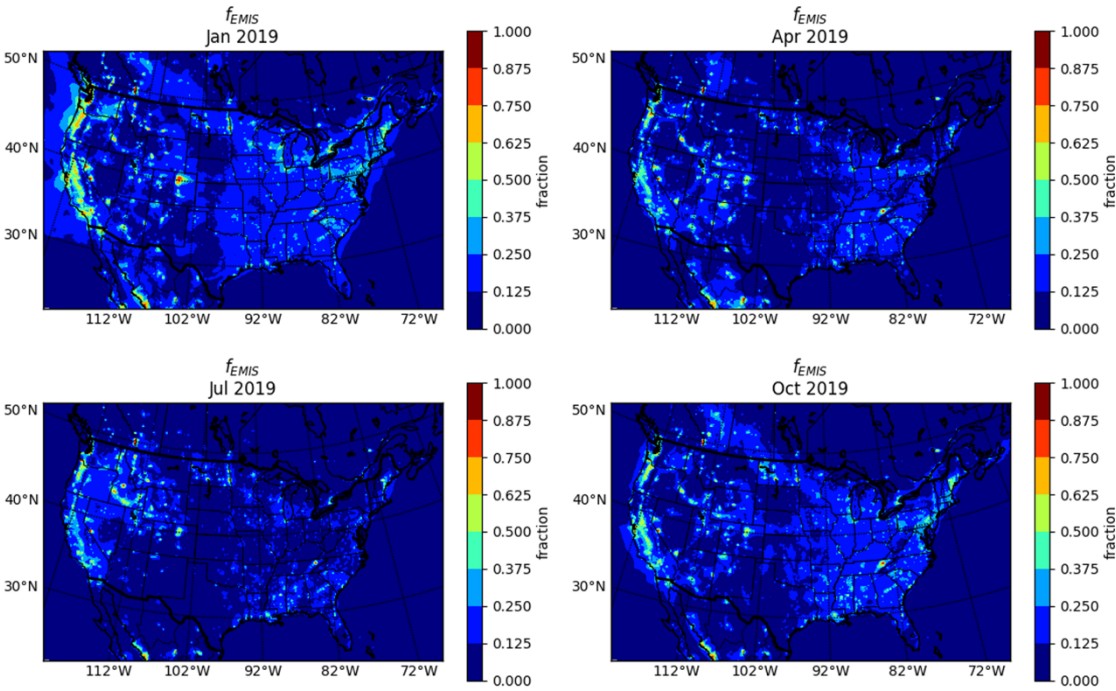

**Fig. 6: The fraction of ASO₄ from primary emission across the contiguous US for the year 2019 in each season (winter: Jan, spring: Apr, summer: July, fall: Oct), based on CMAQ simulation.**

### 3.3 Change in Fractional Annual ASO₄ Formation and Δ¹⁷O(ASO₄) from 2006 to 2019

From 2006 to 2019, the annual Δ¹⁷O(ASO₄) values across the contiguous US showed a consistent increase, highlighting the growing importance of the O₃ pathway in ASO₄ formation (Fig. 7). In the central and eastern US, Δ¹⁷O(ASO₄) values increased by up to 2‰ (Fig. 7), primarily driven by significant reductions in SO₂ emissions, largely attributable to regulatory measures such as the Clean Air Act. These reductions led to lower ASO₄ concentrations, which elevated cloud pH and shifted the ASO₄ formation process toward the O₃ pathway (Fig. S12), resulting in ASO₄ with higher Δ¹⁷O values. Conversely, the western US exhibited only modest increases in Δ¹⁷O(ASO₄), typically less than 1‰ (Fig. 7). This is because the region historically favored O₃-dominated ASO₄ formation due to consistently high O₃ and cloud pH levels (Fig. S13), making the impacts of rising cloud pH and reduced SO₂ emissions less obvious. Changes in H₂O₂ concentrations played a significant role in shaping these trends. In the central and eastern US, slight increases in H₂O₂ concentrations continued to support the H₂O₂ pathway, to a limited extent, even as the O₃ pathway became more dominant (Fig. 7, Fig. S12). In contrast, in the western US, H₂O₂ concentrations decreased slightly, resulting in a slight reduction in $f_{S(IV)+H2O2}$ (Fig. 7). A decrease in $f_{SO2+OH}$ across the domain along with the negligible contributions from other pathways caused a relative increase in $f_{EMIS}$ in these regions (Fig. 7). Between 2006 and 2019, the domain-averaged $f_{SO2+OH}$ decreased by 12.5%, primarily due to the enhanced contribution of the O₃ oxidation

pathway ($f_{f_{S(IV)+O_3}}$) and lower $SO_2$ concentrations under reduced precursor emissions, which together shifted the overall oxidation balance toward aqueous-phase processes. Consistent with these trends, spatial patterns of concentration changes

(Fig. S12) show substantial decreases in $SO_2$ and $ASO_4$, particularly in the eastern US, while $NH_3$ and $ANH_4$ increased, leading to higher cloud pH and favoring $O_3$ oxidation. Meanwhile, primary $ASO_4$ emissions, which do not carry a mass-independent signature and exhibit $\Delta^{17}O$ values close to 0‰, directly added to $ASO_4$ levels and tempered changes in $\Delta^{17}O(ASO_4)$ values (Fig. 7). This dynamic explains why the increases in $\Delta^{17}O(ASO_4)$ values from 2006 to 2019 were smaller in the western US compared to the central and eastern regions.

365

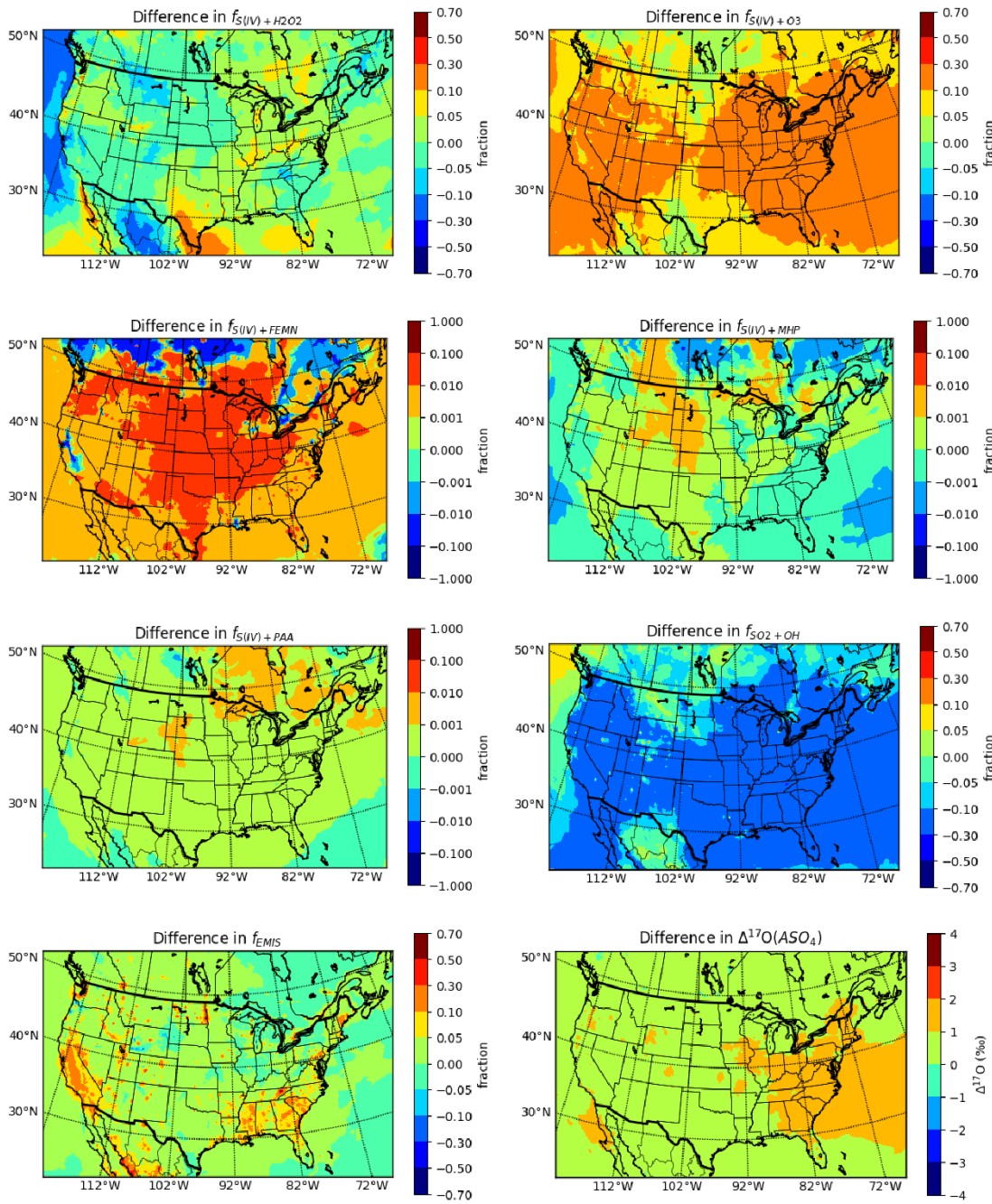

**Fig. 7: The change in the fraction from each ASO₄ formation pathway and Δ¹⁷O values across the contiguous US, from 2006 to 2019, based on CMAQ simulation.**

## 3.4 Comparison of Model $\Delta^{17}O(ASO_4)$ with Observations

The CMAQ simulations of $\Delta^{17}O(ASO_4)$ across the contiguous United States reveal significant insights into atmospheric $ASO_4$ formation over recent decades. However, observations of $\Delta^{17}O(ASO_4)$ in the contiguous US are very limited, with data primarily collected in the late 1990s at La Jolla, CA, and White Mountain Research Station, CA (Lee & Thiemens, 2001), and in the early 2000s at Baton Rouge, LA (Jenkins & Bao, 2006). These historical $\Delta^{17}O(ASO_4)$ data exhibit a range from 0.2‰ to 1.6‰ (Table S1). Due to the predicted change in $ASO_4$ chemistry from 2006 to 2019, the 2006 model simulation was chosen for evaluation against these observations (Fig. 8, 9). The comparison with historical $\Delta^{17}O(ASO_4)$ data is intended as a preliminary evaluation rather than a strict validation, given the temporal mismatch between the available observations (1990s–early 2000s) and the simulation years (2006, 2019). The observed range of 0.2‰ to 1.6‰ provides a useful benchmark for assessing whether the model produces realistic isotopic signatures. However, the limited number and dated nature of these measurements preclude a comprehensive validation of $ASO_4$ chemistry. This further emphasizes the critical need for new $\Delta^{17}O(ASO_4)$ observations within the contiguous United States to enable robust model-observation comparisons.

Generally, the CMAQ model reasonably reproduced $\Delta^{17}O(ASO_4)$ at the Baton Rouge, LA site, with a Root Mean Square Error (RMSE) of 0.20‰ ($n = 17$). This region is characterized by relatively low predicted $\Delta^{17}O(ASO_4)$ values, consistent with high regional $SO_2$ emissions and low cloud water pH that favor $ASO_4$ formation through aqueous $S(IV) + H_2O_2$ reactions. In contrast, the CMAQ-simulated $\Delta^{17}O(ASO_4)$ values tended to be overestimated at the California sites, suggesting possible inaccuracies in representing additional $ASO_4$ production pathways in this region. The La Jolla, CA site had an RMSE of 0.36‰ ($n$=31), while the White Mountain, CA site had a notably higher RMSE of 1.9‰ ($n$=6). Despite the limited number of $\Delta^{17}O(ASO_4)$ observations, a temporal analysis of model simulations versus observations indicates a consistent overprediction during the spring (Fig. 9). The $\Delta^{17}O(ASO_4)$ overestimation in spring could be associated with higher predicted cloud pH during this season, which promotes the $S(IV) + O_3$ oxidation pathway in the model (Fig. S26). The elevated cloud pH may result from increased $NH_3$ emissions, likely related to fertilizer use in surrounding agricultural areas or to underrepresentation of marine boundary layer processes that could influence $ASO_4$ production (Guo et al., 2017; Lim et al., 2022; Zheng et al., 2024; Wang et al., 2025). Given the strong nonlinear pH dependence of $ASO_4$ formation, even moderate $NH_3$ emission biases can produce significant changes in isotopic composition. Future work should include explicit sensitivity simulations to better quantify the coupled effects of $NH_3$, cloud pH, and oxidant chemistry on modeled $\Delta^{17}O(ASO_4)$. While organic acids (e.g., formic, acetic) can locally influence cloud water acidity, their contribution to bulk pH is generally minor relative to the dominant $SO_2$-$H_2SO_4$-$NH_3$ system (Herrmann et al., 2015; Shah et al., 2020; Tsui et al., 2019). Still, future model developments should evaluate their role in regional cloud pH and isotopic composition. Additionally, certain $ASO_4$ formation pathways, such as marine boundary layer chemistry involving $S(IV)$ oxidation by HOX, may not be fully captured, particularly at coastal sites like La Jolla, CA. These reactions can efficiently oxidize $S(IV)$ even under moderately acidic conditions and produce $ASO_4$ with relatively low $\Delta^{17}O$ signatures (Chen et al., 2016; Ishino et al., 2017), which may partly explain the model overestimation at this site. Another

possible factor is the omission of $SO_2$ oxidation via $NO_2$, an emerging multiphase pathway in polluted environments. Such reactions can proceed alongside metal-catalyzed and other aqueous pathways and are anticipated to result in low or near-zero

$\Delta^{17}O$. Sensitivity simulations suggest that this mechanism can enhance $ASO_4$ concentrations by ~0.4-1.2% with a low rate constant and up to 4-20% with a higher rate constant, particularly under polluted, low-oxidant wintertime conditions, when the aqueous S(IV) oxidation by $H_2O_2$ and $O_3$ becomes less efficient (Sarwar et al., 2013), while its overall impact on $\Delta^{17}O(ASO_4)$ is expected to be minor.

Overall, the model-observation comparison of $\Delta^{17}O(ASO_4)$ suggests that CMAQ performs well in more acidic environments but struggles to simulate $ASO_4$ formation under less acidic conditions accurately. However, the limited availability of $\Delta^{17}O(ASO_4)$ observations constrains a more comprehensive evaluation of regional and temporal $ASO_4$ chemistry variations. This highlights the critical need for expanded observational datasets and model refinements to better represent the complex atmospheric $ASO_4$ dynamics. While this study highlights consistent patterns in $ASO_4$ oxidation pathways across the contiguous

US, the evaluation of $\Delta^{17}O(ASO_4)$ remains constrained by the limited and dated nature of available measurements. Expanded and more recent datasets will be essential to validate and extend the findings presented here, particularly to quantify seasonal and regional variability in $ASO_4$ formation.

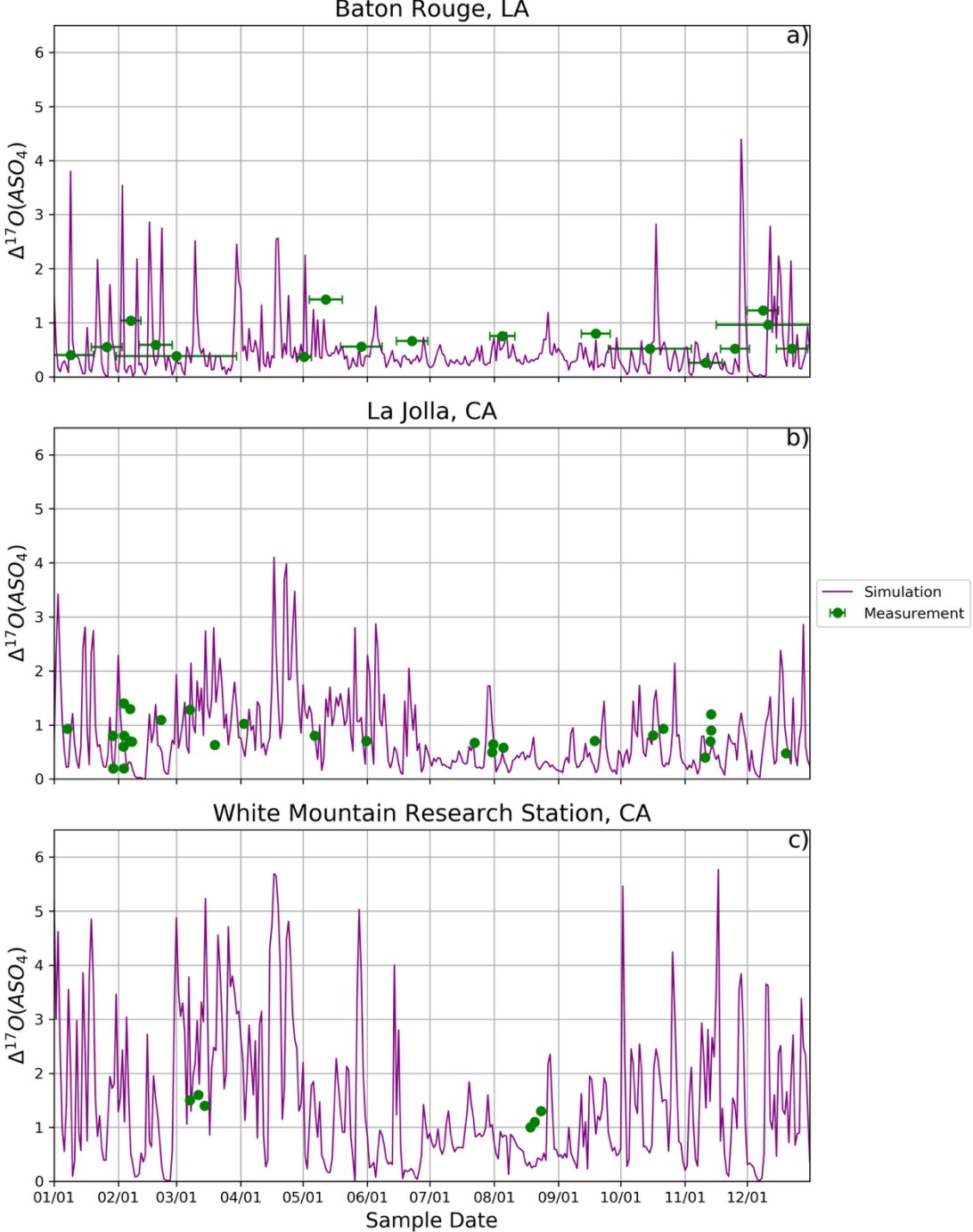

Fig. 8: Temporal variations in $\Delta^{17}O(ASO_4)$ measurements and model simulations at a). Baton Rouge, LA (top); b) La Jolla, CA (middle); and c) White Mountain Research Station, CA (bottom). The x-axis error bars correspond to collection times.

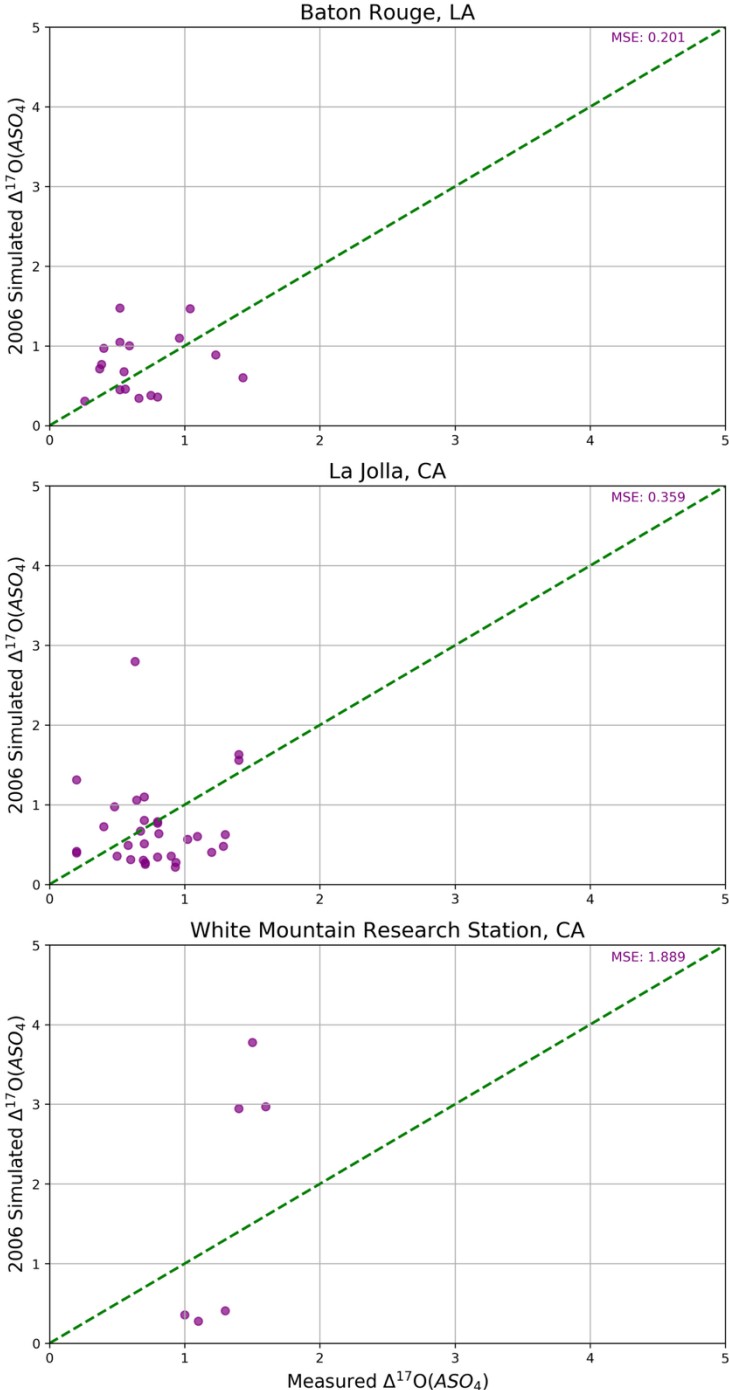

**Fig. 9: Comparison of $\Delta^{17}O(ASO_4)$ measurements and model simulations at La Jolla, CA, White Mountain Research Station, CA, and Baton Rouge, LA from 1996 to 2005.**

## 4 Conclusions

This study modeled $ASO_4$ formation pathways and the $\Delta^{17}O(ASO_4)$ for the contiguous United States using the CMAQ model
for 2006 and 2019. The results reveal distinct seasonal and regional patterns in $ASO_4$ chemistry, strongly influenced by photochemical conditions, emissions of $SO_2$ and $NH_3$, and variations in cloud pH. From 2006 to 2019, significant changes in $ASO_4$ formation dynamics were observed, driven primarily by regulatory-driven reductions in $SO_2$ emissions. These shifts highlight the evolving balance between aqueous-phase oxidation pathways, particularly those driven by $H_2O_2$ and $O_3$.

The reductions in $SO_2$ emissions due to the Clean Air Act resulted in lower cloud water $ASO_4$, which subsequently increased cloud pH. This change shifted $ASO_4$ production toward the $O_3$ pathway, particularly in the eastern US, where the $O_3$ pathway was once limited by lower pH levels in 2006. By 2019, $ASO_4$ formation via $O_3$ oxidation had increased significantly, indicating a more efficient production mechanism under elevated pH conditions. The sub-linear response of $ASO_4$ concentrations to $SO_2$ emission reductions highlights the complexity of $ASO_4$ formation chemistry and the role of co-emitted species, such as $NH_3$,
in modifying pH levels and influencing pathway dominance.

The isotopic signature $\Delta^{17}O(ASO_4)$ serves as a powerful tracer for tracking shifts in $ASO_4$ formation pathways. In regions with limited photochemical activity, such as during winter or in areas with high primary $ASO_4$ emissions, lower $\Delta^{17}O$ values were associated with greater contributions from primary $ASO_4$ emissions. Conversely, higher $\Delta^{17}O$ values reflected an increased
role of the $O_3$ pathway, particularly in regions with elevated cloud pH, reduced $SO_2$ emissions, and higher ozone concentrations.

This work demonstrates a significant and predictable shift in $ASO_4$ chemistry over the study period. The introduction of $\Delta^{17}O(ASO_4)$ as a diagnostic tool for probing $ASO_4$ formation mechanisms provides a novel approach for investigating these
changes. Expanding the measurement of $\Delta^{17}O(ASO_4)$ across diverse regions and time periods will be critical for validating and extending these findings. Future studies should prioritize exploring how changes in atmospheric composition and regulatory measures continue to influence $ASO_4$ chemistry, with a particular focus on understanding the increasing prominence of $O_3$-driven chemistry. This effort will be crucial for enhancing atmospheric models and understanding the implications of $ASO_4$ chemistry on air quality, human health, and climate.


**Code and Data Availability:** The source code for CMAQ version 5.4 is available at
https://github.com/USEPA/CMAQ/tree/5.4 (last access: 1 March 2025). The input datasets for CMAQ simulation are available

at https://cmas-equates.s3.amazonaws.com/index.html#CMAQ_12US1/INPUT/ (last access: 1 March 2025). The in-detail simulation results for $\Delta^{17}O(ASO_4)$ are achieved on Zenodo.org (https://doi.org/10.5281/zenodo.14954960, Fang, 2025).


**Author contributions:** HF and WWW designed the study. HF conducted the model simulations and analysis with input from WWW. HF wrote the manuscript with input from all authors. WWW secured funding.

**Competing Interests:** The contact author has declared that none of the authors have competing interests.


**Acknowledgements:** We thank Kristen Foley for providing the base model input files. We thank Myk Milligan and Nathan Elger and the staff of the Hyperion cluster for helping to install CMAQ, transferring data, and maintaining the computing cluster

**Financial Support:** This research has been supported by NSF AGS (2414561 and 2441725), NSF EPSCOR RII Track-4 (2410015), and USC start-up funds.

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
