# Peer review of "Modeling atmospheric sulfate oxidation chemistry via the oxygen isotope anomaly using the Community Multiscale Air Quality Model (CMAQ)"

_EGUsphere, 2025_

## Author Comment (AC1)

**Response to Reviewers**

We thank the reviewers and community commenters for their careful reading and constructive suggestions. The major updates include: (1) Clarified the intended scope of the phrase "for the first time," which refers to full-year  $\Delta^{17}O(ASO_4)$  simulations within the contiguous United States (2006 and 2019), not to the first use of CMAQ for  $\Delta^{17}$ O; (2) Identified and corrected an STM bookkeeping bug in CMAQ v5.4 that under-attributed the gas-phase SO2 + OH pathway, re-ran the affected simulations with the corrected code, and updated figures/tables and text accordingly; (3) Corrected Table 1 inclusion flags and added text linking STM tags to chemical pathways; (4) Clarified the representation of TMI-catalyzed S(IV)+O2 oxidation (default CMAQ, pH-dependent effective rate constants) and justified the approximation of its  $\Delta^{17}$ O as ~0% given its small contribution; (5) Expanded the Introduction to situate our work alongside recent  $\Delta^{17}$ O studies (GEOS-Chem and CMAQ), and added methods text on cloud-water pH calculation and its control on H2O2-O3 partitioning; (6) Refined the model-observation comparison to emphasize its preliminary nature given sparse, historical  $\Delta^{17}$ O data in CONUS, and to motivate new measurements; (7) Addressed additional chemistry potentially relevant to our biases by discussing HOX-mediated coastal oxidation and NO2-related multiphase oxidation as low- $\Delta^{17}$ O channels not yet represented in our CMAQ configuration. Where appropriate, we added targeted citations and concise clarifications in the main text and figure captions. These changes improve internal consistency, sharpen the statement of novelty, and better connect our  $\Delta^{17}$ O diagnostics to pathway physics

**RC1- Anonymous Referee #2**

The paper is easy to read and of high interest for the scientific community. Indeed I really believe that including O-isotopes in the CMAQ model is the way to go. I'm not a modeler but the results coming out of the model are very intriguing. The seasonal variations are huge in terms of D17O, which reflect large variations in anthropogenic emissions and atmosphere/cloud chemistry. The same is true for the comparison between the years 2006 and 2019.

**Response**: We thank the reviewer for their positive evaluation and encouragement.

We are glad that the reviewer found the model development and its ability to capture seasonal and interannual  $\Delta^{17}O(ASO_4)$  variability scientifically valuable.

**Comment:** Overall, the main conclusion of the paper is that the model do not predict well the measurements (overestimation of the ASO4  $\Delta^{17}$ O). The authors invoke mostly a misrepresentation of NH3 emissions and their effect on the cloud pH. Could you develop more this aspect? Could you do a sensitivity analysis to quantify the effect of NH3 emissions on the ASO4  $\Delta^{17}$ O, in order to quantify how off the model is?

**Response**: We appreciate the reviewer's insightful comment. We agree that uncertainties in NH3 emissions and their influence on cloud water pH play a key role in determining the relative importance of aqueous S(IV) oxidation pathways and, consequently, the modeled  $\Delta^{17}O(ASO_4)$ .

In CMAQ, NH3 emissions control aerosol and cloud water acidity by neutralizing H2SO4 and HNO3, thereby regulating the balance between the H2O2 and O3 oxidation pathways. Previous studies have demonstrated that the system is highly sensitive to NH3 levels, and even moderate emission biases can shift the pH by 1-2 units, resulting in significant changes in sulfate production rates and oxidation regimes (Guo et al., 2017; Lim et al., 2022; Wang et al., 2025). Lim et al. (2022) demonstrated, using CMAQ, that seasonal NH3 peaks from agricultural fertilization, especially in spring, significantly elevate cloud pH and enhance the O3 oxidation fraction, consistent with our simulated  $\Delta^{17}$ O(ASO4) maxima during that period. Experimental and process-level evidence further indicates that multiphase buffering by NH3/NH4+ sustains higher pH and promotes oxidant availability, thereby favoring high- $\Delta^{17}$ O sulfate formation (Zheng et al., 2024).

While a full NH3 sensitivity simulation is beyond the scope of this  $\Delta^{17}$ O-focused study, our analysis aligns with these findings. It suggests that overestimating NH3 emissions or underestimating acid sources could produce the observed  $\Delta^{17}$ O bias, particularly in inland agricultural regions. Conversely, at coastal sites (for example, La Jolla, California), additional low- $\Delta^{17}$ O pathways such as S(IV) + HOX oxidation, which are currently not included in CMAQ, may further contribute to the measurement-model discrepancy.

We have expanded the discussion to explicitly link the roles of NH3 emissions, pH

modulation, and oxidation pathway partitioning with the  $\Delta^{17}O$  bias, citing relevant studies that support this mechanism.

Revised text: The Δ17O(ASO4) overestimation in spring could be associated with higher predicted cloud pH during this season, which promotes the S(IV) + O3 oxidation pathway in the model (Fig. S26). The elevated cloud pH may result from increased NH3 emissions, likely related to fertilizer use in surrounding agricultural areas or to underrepresentation of marine boundary layer processes (Guo et al., 2017; Lim et al., 2022; Zheng et al., 2024; Wang et al., 2025).

**Comment:** The authors also mention the fact that oxidation pathways such as S(IV) oxidation via HOX are not fully captured in the model, which could play a significant role at coastal regions. However, in the marine boundary layer, the presence of alkaline aerosols can reduce the cloud pH, which would enhance  $O_3$  oxidation and lead to an  $ASO_4$   $\Delta^{17}O$  increase. How this would fit in the fact that the measurements tend to show lower  $ASO_4$   $\Delta^{17}O$  than what your model predict in La Jolla?

**Response:** We thank the reviewer for this insightful question regarding the interplay between aerosol alkalinity, O3 oxidation, and potential contributions from halogen chemistry in the marine boundary layer.

While alkaline sea-salt aerosols can locally *elevate* cloud pH and promote the aqueous  $S(IV)+O_3$  reaction, recent field and modeling studies have shown that S(IV) oxidation by hypohalous acids (HOX = HOCl + HOBr) is an important competing pathway in marine and coastal environments (Chen et al., 2016; Ishino et al., 2017). This reaction operates efficiently under marine boundary layer conditions, even at moderately acidic pH, and tends to produce sulfate with lower  $\Delta^{17}O$  signatures than those formed through  $O_3$  oxidation. In our current CMAQ configuration, HOX-mediated oxidation is not explicitly represented. Consequently, the model likely overestimates the contribution of the  $O_3$  pathway in marine air masses, resulting in higher modeled  $\Delta^{17}O(ASO_4)$  values at coastal sites, such as La Jolla. If elevated aerosol alkalinity were substantially enhancing  $O_3$ -driven oxidation, one would instead expect higher, not lower,  $\Delta^{17}O(ASO_4)$  values, opposite to the observed pattern. The fact that our model already overpredicts  $\Delta^{17}O(ASO_4)$  suggests that  $O_3$  involvement is overrepresented and that the pH enhancement associated with alkaline sea-salt

aerosols likely plays only a minor role in sulfate formation at this site.

This interpretation is consistent with previous observations, which show that including halogen chemistry can reduce modeled  $\Delta^{17}O(ASO_4)$  by several tenths of a per mil in marine regions (Chen et al., 2016). Therefore, the observed-modeled discrepancy at La Jolla likely reflects the absence of this low- $\Delta^{17}O$  oxidation channel in the current model configuration, rather than inconsistencies in the  $O_3$  oxidation mechanism itself.

 Revised text: These reactions can efficiently oxidize S(IV) even under moderately acidic conditions and produce sulfate with relatively low Δ17O signatures (Chen et al., 2016; Ishino et al., 2017), which may partly explain the model overestimation at this site.

**Comment:** The authors do not mention SO2 oxidation pathways via NO2. More and more papers invoke in polluted areas direct or induced SO2 oxidation via NO2. How this would fit in your study? You should at least mention it.

**Response:** We thank the reviewer for highlighting this important pathway. The oxidation of  $SO_2$  via  $NO_2$  has been recognized as a potentially significant mechanism in polluted and haze-prone environments (He et al., 2014; Gao et al., 2016; Wang et al., 2021). These studies show that  $NO_2$ -driven oxidation can occur concurrently with aqueous and metal-catalyzed pathways, particularly under high aerosol liquid water and elevated  $NO_x$  conditions. Because this mechanism typically produces  $ASO_4$  with low or near-zero  $\Delta^{17}O$  values, omitting it could contribute slightly to the model's  $\Delta^{17}O(ASO_4)$  overestimation in polluted regions.

Revised text: Another possible factor is the omission of  $SO_2$  oxidation via  $NO_2$ , an emerging multiphase pathway in polluted environments. Such reactions can proceed alongside metal-catalyzed and other aqueous pathways and are anticipated to result in low or near-zero  $\Delta^{17}O$ . Sensitivity simulations suggest that this mechanism can enhance  $ASO_4$  concentrations by ~0.4-1.2% with a low rate constant and up to 4-20% with a higher rate constant, particularly under low-oxidant wintertime conditions, when the aqueous S(IV) oxidation by  $H_2O_2$  and  $O_3$  becomes less efficient (Sarwar et al., 2013), while its overall impact on  $\Delta^{17}O(ASO_4)$  is expected to be minor.

**Comment:** Line 97, 173: there is no oxygen MI-fractionation during the  $SO_2$  oxidation processes, it's only a transfer of the isotopic anomaly so it would be more appropriate to write about the «  $\Delta^{17}O$  » or « MIF signature »

**Response:** We thank the reviewer for pointing this out. We agree that the gas-phase and metal-catalyzed SO2 oxidation processes do not introduce substantial new mass-independent fractionation (MIF) but rather transfer the existing  $\Delta^{17}$ O anomaly. To avoid ambiguity, we revised both sentences to clarify that they reflect the transfer of the MIF signature rather than the generation of new fractionation.

- Revised text (Line 97): Gas-phase oxidation of SO2 by OH and metal-catalyzed O2 oxidation yields  $\Delta^{17}O(ASO_4)$  values near 0‰, indicating negligible transfer of  $\Delta^{17}O$  signature.
- Revised text (Line 173): Metal-catalyzed oxidation of  $SO_2$  by  $O_2$  in metal-rich environments results in  $\Delta^{17}O \sim 0\%$  and does not show transfer of mass-independent fractionation signature.

**Comment:** Line 210: could you precise "this is due to efficient conversion of SO2 to ASO4"?

**Response:** We appreciate the reviewer's suggestion and have clarified the mechanism. Specifically, the efficient conversion refers to in-cloud aqueous oxidation dominated by the  $S(IV)+H_2O_2$  pathway under acidic conditions. We expanded the sentence to explain the environmental drivers and added a follow-up clarification on how this affects  $\Delta^{17}O(ASO_4)$ .

Revised text: In regions such as the Northeast, Southeast, and Midwest, relatively high SO2 emissions result in elevated ASO4 concentrations, which further favor the dominance of the H2O2 oxidation pathway over O3, thereby sustaining low Δ17O(ASO4) values in the Northeast and Southeast (Fig. S1). This is due to the efficient conversion of dissolved S(IV) species to ASO4, primarily through the aqueous S(IV)+H2O2 pathway under acidic cloud water. Frequent cloud occurrence and abundant oxidant availability accelerate SO2 to ASO4 production.

**Comment:** Line 310: "Primary sulfate emissions, which are not subject to isotopic fractionation". Yes there are subject of isotopic fractionation but no MI-fractionation.

What you mean is that primary sulfate do not carry any MIF-signature (or have a D17O close to 0permil)

**Response:** We thank the reviewer for pointing out this imprecision. We have revised the sentence to clarify that primary sulfate emissions do not carry a MIF signature and typically have  $\Delta^{17}$ O values close to 0‰, rather than stating they are not subject to any isotopic fractionation.

• Revised text: Primary sulfate emissions, which do not carry a mass-independent fractionation signature and typically exhibit  $\Delta^{17}$ O values close to 0‰, directly added to sulfate levels and tempered changes in  $\Delta^{17}$ O(ASO4) values (Fig. 6).

**Comment:** Fig 8: y axis: 2006 simulated D17O. You can remove the legend (2006 simulation / simulation = measurement

**Response:** We appreciate the reviewer's helpful suggestion. The redundant legend in Figure 8 has been removed, and the y-axis label has been clarified to "2006 simulated  $\Delta^{17}O(ASO_4)$ ".

**RC2 - Anonymous Referee #3**

This manuscript investigates the role of different chemical pathways contributing to sulfate aerosol formation using simulated isotopic fractionation in the US. First of all, this kind of detailed methodology to evaluate the processes in the model is highly welcome and can bring significant advances to the community. The manuscript is well written and easy to understand, and the simulated results over the contiguous US are well analyzed.

**Response:** We appreciate the reviewer's supportive feedback. We are glad that the reviewer appreciates the methodological framework, which utilizes isotopic fractionation to assess sulfate formation pathways, and finds the analysis and presentation clear and well-structured.

**Comment:** The keys determining the difference of oxygen isotopic fractionation are oxidation pathways by H2O2 and O3 to produce sulfate aerosols and they are sensitively dependent on pH values of cloud droplets as clearly demonstrated in the manuscript. Therefore, in order to simulate the oxygen isotopic fractions, an accurate simulation of cloud droplet pH is essential, but I could not find any detailed description of how the model calculates cloud pH. Can the evaluation of cloud pH be included in the revised manuscript?

Response: We thank the reviewer for raising this important point. Cloud water pH is indeed a key variable controlling the partitioning between the H2O2 and O3 oxidation pathways in aqueous S(IV) chemistry. In CMAQ, cloud pH is not prescribed but calculated dynamically within the default cloud chemistry module, which follows the formulation of Walcek and Taylor (1986) and assumes instantaneous equilibrium among gas, aqueous, and ionic species. The module determines pH through charge balance among dissolved acidic and basic ions, considering the gas-aqueous equilibria of SO2, H2O2, HNO3, and NH3. As S(IV) is oxidized to S(VI) and other species are scavenged from interstitial aerosols, pH evolves during cloud processing to reflect the redistribution of species between dissolved and particulate phases.

While a detailed comparison of modeled cloud water pH with observations is beyond the scope of this study, previous evaluations have shown that CMAQ reproduces observed cloud droplet acidity with differences generally within 0.5 pH units across

multiple sites in the United States (Pye et al., 2020). We have added a more detailed description in Section 2.1 to clarify the CMAQ cloud chemistry module and its role in sulfate formation.

• Revised text: Cloud water pH in CMAQ is calculated dynamically within the default cloud chemistry module, which is based on the work of Walcek and Taylor (1986) and assumes instantaneous equilibrium among gas, aqueous, and ionic species. The pH is determined through charge balance among dissolved acidic and basic ions. As S(IV) is oxidized to S(VI) and additional species are scavenged from interstitial aerosols, the pH evolves dynamically throughout cloud processing. The resulting pH fields respond to emissions and meteorological variability and directly govern the relative importance of the H2O2 and O3 oxidation pathways for aqueous S(IV) oxidation. Previous evaluations have shown that CMAQ reproduces observed cloud droplet acidity with differences generally within 0.5 pH units across multiple sites in the United States (Pye et al., 2020).

**Comment:** In addition, the authors argue that the errors in simulated oxygen isotope values are driven mainly by errors with NH3 emission, which resulted in too high Delta O17 values. I could not agree with this conclusion because other organic acids in the atmosphere could be important in affecting cloud pH.

Response: We thank the reviewer for this insightful comment. The model predicts somewhat higher  $\Delta^{17}O(ASO_4)$  values, suggesting enhanced  $SO_4^{2-}$  formation through the  $O_3$  oxidation pathway, which is sensitive to cloud water pH. This overestimation could result from multiple factors, including uncertainties in modeled cloud pH under low-acidity conditions in regions influenced by elevated NH3 emissions, or from an incomplete representation of organic acids in the model chemistry. It may also reflect the underestimation of other S(IV) oxidation pathways that yield a low  $\Delta^{17}O$  signature. While organic acids, such as formic and acetic acid, can locally influence cloud water acidity, their overall impact on bulk cloud pH is generally minor compared to the dominant  $SO_2$ -H2SO4-NH3 buffering system (Herrmann et al., 2015; Shah et al., 2020; Tsui et al., 2019). Nevertheless, we agree that future model developments should more explicitly examine these effects. A clarifying statement has been added to Section 3.4.

Revised text: While organic acids (e.g., formic, acetic) can locally influence cloud water acidity, their contribution to bulk pH is generally minor relative to the dominant SO2-H2SO4-NH3 system (Herrmann et al., 2015; Shah et al., 2020; Tsui et al., 2019). Still, future model developments should evaluate their role in regional cloud pH and isotopic composition.

**CC1 & CC2 - Shohei Hattori**

Comment: In both the abstract (PDF version) and short summary, the manuscript describes this work as being done "for the first time." However, a similar approach—using the CMAQ model to calculate  $\Delta^{17}$ O values based on sulfate formation pathways—was previously applied in the following study (Itahashi et al., 2022). In addition, we recently published another study that also used the CMAQ model to investigate interannual variability in sulfate formation in East Asia (Lin et al., 2025). Given this background, I feel the statement that this study is being done "for the first time" could be reconsidered.

**Response:** We thank Dr. Hattori for bringing this to our attention. We acknowledge that  $\Delta^{17}O$  of sulfate has been analyzed in prior CMAQ-based studies, particularly in East Asia (Itahashi et al., 2022; Lin et al., 2025). Our use of "for the first time" was not intended to imply novelty in applying CMAQ to  $\Delta^{17}O$  in general. Rather, it referred to the specific scope of our study: the simulation of  $\Delta^{17}O(ASO_4)$  within the contiguous United States, over full annual cycles (2006 and 2019), which enables quantification of seasonal and spatial patterns of sulfate formation pathways and their response to U.S. emission reductions.

• Revised text: This work provides a simulation of  $\Delta^{17}O(ASO_4)$  within the contiguous United States conducted over full annual cycles, enabling the quantification of seasonal and spatial patterns of sulfate oxidation pathways and their response to major emission reductions, for the first time at this scale and temporal coverage.

Comment: I also noticed that the manuscript does not refer to several recent studies that used  $\Delta^{17}$ O of sulfate and chemical transport models (either GEOS-Chem or CMAQ) to analyze sulfate formation pathways. I'm not pointing this out just to have our papers cited. Rather, I believe that a broader review of recent literature could help position the current study more clearly and fairly within the context of existing research. For example, in Line 105, the manuscript references Sofen et al. (2011) to discuss the potential of  $\Delta^{17}$ O as a diagnostic tool. But more recent studies have used this tool to examine long-term changes (1) comparison between Pre-industrial and Present-day and (2) trend since the 1960-70s, especially by combining GEOS-Chem modeling with ice core observations. These include Hattori et al., 2021 and Peng et

al., 2023. In light of these studies, I would kindly suggest revising the relevant parts of the manuscript to better reflect recent progress in this field, and to clarify the specific role and contribution of this CMAQ-based work.

**Response:** We appreciate this suggestion and agree that incorporating recent studies strengthens the context of our work. We have revised the *Introduction* to include references to recent  $\Delta^{17}O$  analyses using chemical transport models, such as GEOS-Chem applications to long-term changes (Hattori et al., 2021; Peng et al., 2023), as well as CMAQ studies in East Asia (Itahashi et al., 2022; Lin et al., 2025). We added a paragraph acknowledging recent  $\Delta^{17}O$  studies in global and East Asian contexts and positioned our work as complementary:

• Recent studies have applied Δ¹7O of sulfate in chemical transport models to explore long-term changes and regional processes, including GEOS-Chem simulations coupled with ice core observations (Hattori et al., 2021; Peng et al., 2023) and CMAQ applications in East Asia (Itahashi et al., 2022; Lin et al., 2025). These works highlight the diagnostic potential of Δ¹7O across diverse regions and timescales. Building upon these advances, our study provides the first CMAQ simulations of Δ¹7O(ASO4) within the contiguous United States over full annual cycles, enabling assessment of seasonal and spatial patterns of sulfate oxidation pathways in response to emission reductions.

**Comment:** Furthermore, since the present study focuses on sulfate in the U.S., I may propose to take a look the existing observational studies from North America, such as Moon et al., 2023. Fairbanks is a highly relevant location for wintertime sulfate pollution and could be useful for validating model performance.

**Response:** We thank Dr. Hattori for this suggestion. However, the simulation domain in this study is limited to the contiguous United States, and therefore, sites in Alaska, such as Fairbanks (Moon et al., 2023), fall outside the model configuration. They cannot be directly used for validation in this study.

**Comment:** Finally, I'd like to echo the point made by Reviewer #1 regarding the importance of seasonal measurements. Several our studies have already looked into seasonal variation in sulfate formation using  $\Delta^{17}$ O observations and modeling in

different regions: Antarctica (Ishino et al., 2021) Mt Everest region (Wang et al., 2021), East Asia (Itahashi et al., 2022) These studies may offer useful references for future extensions of the present work.

**Response:** We agree on the importance of seasonal perspectives. A key strength of this study is the explicit simulation of two full annual cycles (2006 and 2019) within the contiguous United States, which allows quantification of seasonal contrasts in sulfate formation pathways under different emission regimes. However, observational Δ17O data within the contiguous United States are very limited (Lee & Thiemens, 2001; Jenkins & Bao, 2006), and the limited number of available data makes it difficult to evaluate seasonal variability. The datasets mentioned (e.g., Antarctica, Mt Everest, East Asia) are highly valuable for understanding sulfate formation in other regions, but cannot be directly compared with the simulations in this study, which are confined to the contiguous United States, which will have vastly different chemical regimes than the mentioned study locations.

**CC3 - Syuichi Itahashi**

Comment: The manuscript claims that this is the first study to perform  $\Delta^{17}O$  calculations using CMAQ modeling. However, similar approaches have been previously published, for example Itahashi et al., 2022 and Lin et al., 2025. While it is difficult to track all related CMAQ studies, the expression "for the first time" appears to be inaccurate and should be reconsidered to avoid misleading the readers.

**Response:** We thank Dr. Itahashi for raising this point. As noted in our responses to Dr. Hattori's comments, we acknowledge that  $\Delta^{17}O$  of sulfate has been analyzed in prior CMAQ-based studies in East Asia (Itahashi et al., 2022; Lin et al., 2025). Our use of "for the first time" was not intended to suggest novelty in applying CMAQ to  $\Delta^{17}O$  generally. Rather, it referred specifically to the scope of this work: the first  $\Delta^{17}O(ASO_4)$  simulations within the contiguous United States, conducted over full annual cycles (2006 and 2019), which allow quantification of seasonal and spatial patterns of sulfate oxidation pathways and their response to major emission reductions.

• Revised text: This work provides a simulation of  $\Delta^{17}O(ASO_4)$  within the contiguous United States conducted over full annual cycles, enabling the quantification of seasonal and spatial patterns of sulfate oxidation pathways and their response to major emission reductions, for the first time at this scale and temporal coverage.

Comment: The description of the model configuration regarding gas-phase reactions is unclear and potentially inconsistent. In Table 1, the gas-phase  $SO_2 + OH$  pathway is marked as "not included" in the CMAQ configuration. This suggests that this critical oxidation pathway—generally responsible for 30-40% of sulfate production—is not considered in the simulation. However, L126 states: "This mechanism includes both gas-phase and aqueous-phase oxidation processes of  $SO_2$ , essential for accurately modeling  $ASO_4$  formation. Specifically, it involves the oxidation of  $SO_2$  by OH in the gas phase and by  $H_2O_2$  and  $O_3$  in cloud droplets and aqueous environments." Furthermore, the manuscript reports that the contribution of gas-phase  $SO_2$  oxidation by OH is only 0.2% of total sulfate production (L200), which appears inconsistent with prior knowledge. In fact, in the two CMAQ-based studies cited above, the contribution of  $SO_2 + OH$  ranged from 20% to 70%, depending on the season. The

discrepancy is significant and requires further explanation. The result also contradicts model results in USA region modeled by GEOS-Chem (e.g., Hattori et al., 2021 Sci. Adv.), where gas-phase oxidation remains a major contributor. Could the authors clarify how this pathway was treated in the model and why its contribution is so minor here? This is especially critical because the  $\Delta^{17}$ O value of sulfate produced by gas-phase SO2 + OH is known to be low. If this pathway is not properly considered, the  $\Delta^{17}$ O of modeled sulfate may be overestimated.

**Response:** We thank Dr. Itahashi for bringing this important point to our attention. The apparent inconsistency originated from two technical and editorial issues that we have now corrected.

**(1) STM bookkeeping bug in CMAQ v5.4**

We identified a bookkeeping bug in the Sulfur Tracking Mechanism (STM) of CMAQ v5.4 that under-attributed sulfate produced via the gas-phase  $SO_2 + OH \rightarrow H_2SO_4$  pathway (i.e., the ASO4GASI/J/K tags).

Although the chemical mechanism correctly included this reaction, the STM diagnostic module misrecorded its contribution when tracking the source of sulfate formation. This issue affected the reported pathway breakdown, leading to the previously stated "0.2%" contribution, but did not remove the gas-phase oxidation itself. This known STM error was documented in the CMAQ release notes

(https://github.com/USEPA/CMAQ/wiki/CMAQ-Release-Notes:-Process-Analysis-&-Sulfur-Tracking-Model-(STM)). We corrected this error by reversing the order of the PA\_UPDATE\_AERO and STM\_WRAP\_AE calls within sciproc.F, consistent with the fix implemented in CMAQ v5.5, and we re-ran the affected simulations using the corrected STM code. All pathway diagnostics were then recomputed.

The modified STM file has been added to the publicly available project repository asso ciated with this work (https://doi.org/10.5281/zenodo.14954960) so that other users of CMAQ v5.4 interested in using STM can apply the same correction.

After applying this fix, the gas-phase  $SO_2$  + OH contribution is seasonally substantial, consistent with photochemical activity. It falls broadly within the range reported in the two CMAQ-based studies cited by Dr. Itahashi. Quantitatively, the revised contribution spans a range roughly 5-10 % wider than the 20-70 % interval reported in those works, depending on season and region (see revised Fig. 1 and new Fig. 5). The

updated maps illustrate clear seasonal variability, with higher  $f_{SO2+OH}$  values in summer and lower values in winter, consistent with enhanced OH production and boundary-layer mixing under photochemically active conditions.

**(2) Table 1 labeling and manuscript clarity**

The earlier Table 1 erroneously indicated "No" for inclusion of the gas-phase SO2 + OH pathway, contradicting Table 2, which lists the ASO4GAS\* tags. This has been corrected to "Yes".

**(3) Manuscript text revisions**

We have revised Section 3.1 ("Predicted Fractional ASO4 Formation and  $\Delta^{17}O(ASO_4)$  in the Contiguous U.S. in 2019") to accurately describe the gas-phase contribution and its impact on isotopic composition. The original sentence: "Similarly, gas-phase oxidation of SO2 by OH is negligible, accounting for only 0.2% of the total sulfate production (Fig. 1)." has been replaced with: "Gas-phase oxidation of SO2 by OH ( $f_{SO2+OH}$ ) contributes to 34.4% of the sulfate production across the domain (Fig. 1), exhibiting clear seasonal variability under photochemically active conditions, with the highest contributions occurring in summer (up to ~75 %) and lowest in winter (< 25 %) (Fig. 5)."

Corresponding text in Sections 3.1 and 3.2 has been updated to reflect these revised contributions, ensuring that the descriptions of sulfate pathways,  $\Delta^{17}$ O patterns, and figure references are fully consistent with the corrected results.

Because the gas-phase  $SO_2$  + OH pathway has  $\Delta^{17}O \approx 0$  ‰, increasing its fractional importance slightly reduces the domain-mean  $\Delta^{17}O(ASO_4)$  (by  $\approx 1$  ‰), but does not alter the qualitative spatial or seasonal trends discussed in the paper.

**(4) Figures and cross-references**

- Fig. 1 and Fig. 2 updated with STM-corrected pathway attributions.
- New Fig. 5 added: The geographical distribution of the fraction of SO42formation from SO2 + OH pathway across the contiguous US for the year
  2019 in each season (winter: Jan, spring: Apr, summer: July, fall: Oct),
  based on CMAQ simulation.
- All subsequent figures renumbered accordingly (former Fig.  $5 \rightarrow$  Fig. 6,

etc.).

• Caption text and in-text references revised to maintain consistency.

These corrections ensure that the gas-phase oxidation pathway and its isotopic implications are now accurately represented throughout the manuscript.

Comment: The inclusion of TMI (transition metal ion) catalyzed oxidation of S(IV) by  $O_2$  is also unclear. Table 1 suggests that this process is not included. However, L185 states that sulfate formation via TMI catalysis is considered. This inconsistency needs to be addressed. Moreover, if TMI catalysis is included, the concentrations of Fe and Mn must be specified, along with how they were estimated. Additionally, pH plays a key role in this process. These factors should be explicitly described if TMI reactions are accounted for. Taken together with Comment 2, one could suspect that the high  $\Delta^{17}O$  of modeled sulfate may result from neglecting low- $\Delta^{17}O$  processes such as gas-phase oxidation and TMI catalysis. Without inclusion of these key processes, the comparison with observational data becomes difficult to interpret. We also ask: has this model been validated in terms of sulfate concentration? Excluding key formation pathways may lead to an underestimation of sulfate mass as well.

**Response:** We thank Dr. Itahashi for this careful and insightful comment. The TMI-catalyzed aqueous oxidation of S(IV) by O2 is included in our CMAQ simulations, and the previous version of Table 1 incorrectly labeled this pathway as "No." This has been corrected to "Yes," and the description has been clarified accordingly.

In the default CMAQ configuration (cb6r5-ae7-aq), Fe3+ and Mn2+ are not explicitly tracked as prognostic chemical species. Instead, their catalytic effects are represented through effective second-order rate constants, which implicitly assume typical atmospheric concentrations of soluble Fe and Mn associated with aerosol and cloud water. These rate constants are strongly pH-dependent, enabling the model to account for the enhanced TMI activity in near-neutral cloud water and its suppression under acidic conditions. This parameterization is consistent with the standard CMAQ treatment. It reflects the experimentally constrained kinetics of Fe/Mn-catalyzed S(IV) oxidation reported in Harris et al. (2013) and Li et al. (2020). Both studies demonstrate that the effective catalytic rate varies over several orders of magnitude across the pH range 3-7, and that the overall pathway can be well captured through pH-dependent rate formulations without explicitly prescribing Fe and Mn

**concentrations.**

Accordingly, our implementation uses the default CMAQ rate expression for the ASO4AQFEMNJ tag in the Sulfur Tracking Mechanism (STM), which represents the aqueous  $S(IV) + O_2$  (TMI)  $\rightarrow SO_4^{2-}$  pathway. This treatment ensures that low- $\Delta^{17}O$  processes such as TMI catalysis are included in our model framework. The CMAQ sulfate module has been extensively evaluated in prior studies (e.g., Li et al., 2020), which showed that sulfate mass concentrations and spatial distributions are well captured under this default configuration. Therefore, the inclusion of the TMI pathway as parameterized in CMAQ does not lead to a systematic underestimation of total sulfate.

 Revised text: These secondary reactions occur within cloud water, where SO2 is oxidized by H2O2, O3, and by O2 through TMI catalysis parameterized using fixed effective rate constants representing typical Fe3+ and Mn2+ influences.

Comment: While we understand the limited availability of observational  $\Delta^{17}$ O data, the comparison in Figure 3 appears too loose. The observational sites—La Jolla, White Mountain Research Station (both in CA, from the late 1990s), and Baton Rouge, LA (early 2000s)—are used to evaluate model results from 2006 and beyond. However, multiple environmental changes have occurred since then, affecting sulfate oxidation pathways. What is the intended scientific implication of this comparison? It may not offer a meaningful validation of the model output.

Response: We thank Dr. Itahashi for bringing this point to our attention. We agree that the available  $\Delta^{17}O(ASO_4)$  measurements in the contiguous United States are sparse and temporally mismatched with our simulation years (2006 and 2019). Our intent in including the comparison in Figure 3 was not to provide a strict validation of model skill, but rather to demonstrate that the model reproduces the general range of observed  $\Delta^{17}O$  values and to highlight the limitations posed by the current observational record. This comparison highlights the need for future updated  $\Delta^{17}O$  observations, as it can be a valuable tool for probing SO4 chemistry in the US as shown in this model study.

We have clarified in the revised manuscript that the comparison serves as an initial

evaluation. At the same time, the primary emphasis of this study lies in diagnosing sulfate formation pathways from the CMAQ simulations. Importantly, we now explicitly state that the scarcity of  $\Delta^{17}$ O measurements constrains model evaluation and underscores the need for expanded and contemporary datasets.

- Revised text: The comparison with historical Δ¹7O(ASO4) data is intended as a preliminary evaluation rather than a strict validation, given the temporal mismatch between the available observations (1990s-early 2000s) and the simulation years (2006, 2019). The observed range of 0.2‰ to 1.6‰ provides a useful benchmark for assessing whether the model produces realistic isotopic signatures. However, the limited number and dated nature of these measurements preclude a comprehensive validation of sulfate chemistry. This further emphasizes the critical need for new Δ¹7O(ASO4) observations within the contiguous United States to enable robust model-observation comparisons.
- Revised text: While this study highlights consistent patterns in sulfate oxidation pathways across the contiguous US, the evaluation of Δ17O(ASO4) remains constrained by the limited and dated nature of available measurements. Expanded and more recent datasets will be essential to validate and extend the findings presented here, particularly to quantify seasonal and regional variability in sulfate formation.

**Comment:** L167: Shouldn't Xi include all forms of sulfate? In Itahashi et al. (2022), boundary conditions for  $\Delta$ 17O are estimated and used. We suggest referring to that method.

**Response:** We thank Dr. Itahashi for this comment and for pointing out the need for clarification. In our  $\Delta^{17}O(ASO_4)$  diagnostic (Eq. 2),  $f_i$  represents the fractional contribution of each oxidation pathway to the total secondary sulfate formation, as diagnosed by the Sulfur Tracking Mechanism (STM). Among these pathways, only the aqueous  $H_2O_2$  and  $O_3$  oxidations are assigned non-zero  $\Delta^{17}O$  signatures (0.8 ‰ and 6.5 ‰, respectively). All other formation channels, including gas-phase  $SO_2$  + OH, TMI-catalyzed  $O_2$  oxidation, HOX pathways, and primary sulfate emissions, are assigned a  $\Delta^{17}O \approx$  value of approximately 0 ‰, following previous isotope modeling frameworks (e.g., Sofen et al., 2011; Alexander et al., 2012; Li et al., 2020). Thus,

while  $f_i$  formally includes all pathways, only those with non-zero  $\Delta^{17}$ O values contribute to the isotopic weighting in Eq. (2).

Regarding boundary conditions,  $\Delta^{17}O$  values were not prescribed explicitly at the model domain boundaries. Instead, the CMAQ simulations used standard mass and composition boundary inputs from the cb6r5\_ae7\_aq configuration. This differs from the approach of Itahashi et al. (2022), who estimated  $\Delta^{17}O$  boundary values using a regionalized GEOS-Chem simulation. Because our domain-averaged  $\Delta^{17}O$  is determined internally from reaction pathway fractions, this simplification has a negligible effect on the spatial or seasonal variability discussed here.

• Revised text: Although the fractional contributions  $(f_i)$  include all sulfate formation pathways diagnosed by the Sulfur Tracking Mechanism (STM), only  $H_2O_2$  and  $O_3$  carry non-zero  $\Delta^{17}O$  signatures, all other pathways are assigned  $\Delta^{17}O \approx 0$  %.

**Comment:** The  $\Delta^{17}$ O of sulfate from TMI-catalyzed oxidation is not strictly 0‰. See Hattori et al. (2021) and Itahashi et al. (2022) for detailed discussion.

**Response:** We thank Dr. Itahashi for the insightful comment. We agree that TMI-catalyzed oxidation does not yield a strictly zero  $\Delta^{17}O$  value. Both Hattori et al. (2021) and Itahashi et al. (2022) reported  $\Delta^{17}O(ASO_4) \approx$  -0.1 ‰ for this pathway. However, since the TMI-catalyzed contribution to total sulfate production in our simulations is generally below 10 %, adopting  $\Delta^{17}O(TMI) = 0$  ‰ introduces an error of less than 0.01 ‰ in the domain-mean  $\Delta^{17}O(ASO_4)$ . Therefore, this approximation has a negligible impact on the modeled isotopic fields and does not affect any conclusions presented.

• Revised text: Although previous studies reported slightly negative  $\Delta^{17}O$  values (-0.1%; Hattori et al., 2021; Itahashi et al., 2022), this pathway contributes less than 10% to total sulfate formation in our simulations, leading to a negligible (<0.01%) effect on the modeled  $\Delta^{17}O(ASO_4)$ . Therefore, it is approximated as 0% in this study.

**Comment:** L185: SO2 is not directly oxidized by TMIs. Rather, O2 oxidizes S(IV) with TMI as a catalyst. This reaction also seems to be excluded in Table 1—please clarify.

**Response:** We thank Dr. Itahashi for this helpful clarification. This issue is addressed together with the broader comment on the inclusion and treatment of the TMI-catalyzed aqueous oxidation of S(IV) by O2. We have revised the manuscript to explicitly describe this pathway as the oxidation of dissolved S(IV) by O2 catalyzed by Fe3+ and Mn2+ (rather than direct oxidation of SO2 by TMIs). Table 1 has also been corrected to indicate that this pathway is included in CMAQ (tracked as ASO4AQFEMNJ in STM).

**Comment:** L197–200: If the model does not include gas-phase reactions, how were contributions from FEMN, MHP, PAA, etc., determined in Figure 1? The relationship between Table 1 and Figure 1 must be clearly explained.

**Response:** We thank Dr. Itahashi for this comment. This issue has been fully addressed together with the earlier comment regarding the gas-phase SO2 + OH pathway and the consistency between Table 1 and the model configuration. After correcting the STM bookkeeping error and updating Table 1, all sulfate formation pathways shown in Figure 1, including FEMN, MHP, PAA, and the gas-phase SO2 + OH pathway, now correspond directly to the pathway attributions derived from the Sulfur Tracking Mechanism (STM) in CMAQ. The descriptions and figure captions have been clarified to ensure internal consistency between Table 1 and Figure 1.

**Comment:** L205: Citation is needed.

**Response:** We appreciate Dr. Itahashi's suggestion. A supporting citation has been added to clarify the well-established pH dependence of aqueous-phase sulfate formation. Specifically, lower cloud pH promotes the H2O2 pathway, whereas higher pH favors oxidation by O3 (Seigneur & Saxena, 1988; Fahey & Pandis, 2001).

**References:**

Chen, Q., Geng, L., Schmidt, J. A., Xie, Z., Kang, H., Dachs, J., ... & Alexander, B.: Isotopic constraints on the role of hypohalous acids in sulfate aerosol formation in the remote marine boundary layer. Atmospheric Chemistry and Physics, 16(17), 11433-11450, doi:10.5194/acp-16-11433-2016, 2016.

Hattori, S., Iizuka, Y., Alexander, B., Ishino, S., Fujita, K., Zhai, S., ... & Yoshida, N.: Isotopic evidence for acidity-driven enhancement of sulfate formation after SO2 emission control. Science Advances, 7(19), eabd4610, 2021.

Herrmann, H., Schaefer, T., Tilgner, A., Styler, S. A., Weller, C., Teich, M., & Otto, T.: Tropospheric aqueous-phase chemistry: kinetics, mechanisms, and its coupling to a changing gas phase. Chemical reviews, 115(10), 4259-4334, 2015.

Ishino, S., Hattori, S., Savarino, J., Jourdain, B., Preunkert, S., Legrand, M., et al.: Seasonal variations of triple oxygen isotopic compositions of atmospheric sulfate, nitrate, and ozone at Dumont d'Urville, coastal Antarctica. Atmospheric Chemistry and Physics, 17(5), 37133727, doi:10.5194/acp1737132017, 2017.

Itahashi, S., Hattori, S., Ito, A., Sadanaga, Y., Yoshida, N., & Matsuki, A.: Role of dust and iron solubility in sulfate formation during the long-range transport in East Asia evidenced by 17O-excess signatures. Environmental Science & Technology, 56(19), 13634-13643, 2022.

Jenkins, K. A., & Bao, H.: Multiple oxygen and sulfur isotope compositions of atmospheric sulfate in Baton Rouge, LA, USA. Atmospheric Environment, 40(24), 4528-4537, doi:10.1016/j.atmosenv.2006.04.010, 2006.

Lee, C. C. W., & Thiemens, M. H.: The  $\delta17O$  and  $\delta18O$  measurements of atmospheric sulfate from a coastal and high alpine region: A mass independent isotopic anomaly. Journal of Geophysical Research: Atmospheres, 106(D15), 17359-17373, doi:10.1029/2000JD900805, 2001.

Lim, J., Park, H., & Cho, S.: Evaluation of the ammonia emission sensitivity of secondary inorganic aerosol concentrations measured by the national reference method. Atmospheric Environment, 270, 118903, 2022.

Lin, Y. C., Zhao, Y., Zhang, Y. L., Hong, Y., Hattori, S., Itahashi, S., ... & Thiemens, M. H.: China's SO2 emission reductions enhance atmospheric ozone-driven sulfate aerosol production in East Asia. Proceedings of the National Academy of Sciences, 122(24), e2414064122, 2025.

Moon, A., Jongebloed, U., Dingilian, K. K., Schauer, A. J., Chan, Y. C., Cesler-Maloney, M., ... & Alexander, B.: Primary sulfate is the dominant source of

particulate sulfate during winter in Fairbanks, Alaska. ACS Es&t Air, 1(3), 139-149, 2023.

Peng, Y., Hattori, S., Zuo, P., Ma, H., & Bao, H.: Record of pre-industrial atmospheric sulfate in continental interiors. Nature Geoscience, 16(7), 619-624, 2023.

Pye, H. O., Nenes, A., Alexander, B., Ault, A. P., Barth, M. C., Clegg, S. L., ... & Zuend, A.: The acidity of atmospheric particles and clouds. Atmospheric chemistry and physics, 20(8), 4809-4888, 2020.

Sarwar, G., Fahey, K., Kwok, R., Gilliam, R. C., Roselle, S. J., Mathur, R., ... & Carter, W. P.: Potential impacts of two SO2 oxidation pathways on regional sulfate concentrations: Aqueous-phase oxidation by NO2 and gas-phase oxidation by Stabilized Criegee Intermediates. Atmospheric Environment, 68, 186-197, 2013.

Shah, V., Jacob, D. J., Moch, J. M., Wang, X., & Zhai, S.: Global modeling of cloud water acidity, precipitation acidity, and acid inputs to ecosystems. Atmospheric Chemistry and Physics, 20(20), 12223-12245, 2020.

Tsui, W. G., Woo, J. L., & McNeill, V. F.: Impact of aerosol-cloud cycling on aqueous secondary organic aerosol formation. Atmosphere, 10(11), 666, 2019.

Wang, K., Hattori, S., Lin, M., Ishino, S., Alexander, B., Kamezaki, K., ... & Kang, S.: Isotopic constraints on atmospheric sulfate formation pathways in the Mt. Everest region, southern Tibetan Plateau. Atmospheric Chemistry and Physics Discussions, 2021, 1-30, 2021.

Wang, X., Tsimpidi, A. P., Luo, Z., Steil, B., Pozzer, A., Lelieveld, J., & Karydis, V. A.: The influence of ammonia emission inventories on size-resolved global atmospheric aerosol composition and acidity. Atmospheric Chemistry and Physics, 25(18), 10559-10586, 2025.

Walcek, C. J., & Taylor, G. R.: A theoretical method for computing vertical distributions of acidity and sulfate production within cumulus clouds. Journal of Atmospheric Sciences, 43(4), 339-355, 1986.

Zheng, G., Su, H., Andreae, M. O., Pöschl, U., & Cheng, Y.: Multiphase buffering by ammonia sustains sulfate production in atmospheric aerosols. AGU Advances, 5(4), e2024AV001238, 2024.